# Sleep fMRI with simultaneous electrophysiology at 9.4 T in male mice

Yalin Yu[1,2,9], Yue Qiu [3,9], Gen Li[4], Kaiwei Zhang[1], Binshi Bo[1], Mengchao Pei[1], Jingjing Ye[5], Garth J. Thompson [5], Jing Cang[3], Fang Fang[3,6], Yanqiu Feng [7], Xiaojie Duan [4] ✉, Chuanjun Tong [1,7] ✉ & Zhifeng Liang [1,8] ✉

Sleep is ubiquitous and essential, but its mechanisms remain unclear. Studies in animals and humans have provided insights of sleep at vastly different spatiotemporal scales. However, challenges remain to integrate local and global information of sleep. Therefore, we developed sleep fMRI based on simultaneous electrophysiology at 9.4 T in male mice. Optimized unanesthetized mouse fMRI setup allowed manifestation of NREM and REM sleep, and a large sleep fMRI dataset was collected and openly accessible. State dependent global patterns were revealed, and state transitions were found to be global, asymmetrical and sequential, which can be predicted up to 17.8 s using LSTM models. Importantly, sleep fMRI with hippocampal recording revealed potentiated sharp-wave ripple triggered global patterns during NREM than awake state, potentially attributable to co-occurrence of spindle events. To conclude, we established mouse sleep fMRI with simultaneous electrophysiology, and demonstrated its capability by revealing global dynamics of state transitions and neural events.

Sleep is generally considered a tightly regulated whole-brain phenomenon, and its role for our cognition and heath is, from our own daily experience, of great significance. In recent years, studies have identified neural circuits regulating transition and stability of awake-sleep cycle[1,2]. The ascending arousal system[3], including monoaminergic, cholinergic, and peptidergic systems, are involved in promoting or sustaining wakefulness. Genetically defined cell populations in the preoptic area (POA), basal forebrain (BF), brainstem, and cortex, have been identified as non-rapid eye movement (NREM) promoting cells[2]. And neurons in pedunculopontine tegmentum (PPT) and laterodorsal tegmentum (LDT) are involved in rapid eye movement (REM) sleep generation[2]. In addition, the sleep/wake states are usually accompanied by distinct neural events which result from the synchronous activities of neural circuits, for example, spindles or slow waves in NREM sleep[4,5] and sharp wave ripples (SWRs)[6] in both quiet wakefulness and NREM sleep. Previous studies have shown that these events play important roles in sleep architecture, synaptic plasticity and memory consolidation[6,7], among many others. Circuitry level approaches in animals have provided detailed understanding of neural mechanisms during awake-sleep cycle, but techniques providing macroscopic view are also needed for systematical examination of sleep, a fundamentally global phenomenon.

Non-invasive brain mapping tools such as functional magnetic resonance imaging (fMRI), electroencephalogram (EEG), positron emission computed tomography (PET) have provided the whole brain insights into sleep. Previous PET studies showed that the

[1]Institute of Neuroscience, CAS Key Laboratory of Primate Neurobiology, Center for Excellence in Brain Science and Intelligence Technology, Chinese Academy of Sciences, Shanghai, China. [2]University of Chinese Academy of Sciences, Beijing, China. [3]Department of Anesthesia, Zhongshan Hospital, Fudan University, Shanghai, China. [4]Department of Biomedical Engineering, College of Future Technology, Academy for Advanced Interdisciplinary Studies, National Biomedical Imaging Centre, Peking University, Beijing, China. [5]iHuman Institute, ShanghaiTech University, Shanghai, China. [6]The Central Hospital of Xuhui District, Shanghai, China. [7]School of Biomedical Engineering, Southern Medical University, Guangzhou, China. [8]Shanghai Center for Brain Science and Brain-Inspired Intelligence Technology, Shanghai, China. [9]These authors contributed equally: Yalin Yu, Yue Qiu. ✉e-mail: xjduan@pku.edu.cn; trangetung@163.com; zliang@ion.ac.cn

descent from wakefulness to NREM sleep was accompanied by global or regional reductions in cerebral blood flow (CBF), oxygen metabolism and glucose metabolism[8,9]. Blood-oxygen-level-dependent (BOLD) fMRI studies also demonstrated widespread hemodynamic alterations during awake-NREM cycle[10]. Coherent patterns of slow and large-amplitude oscillating electrophysiological, hemodynamic, and cerebral-spinal fluid (CSF) dynamics were found during NREM sleep[11], and BOLD signals further exhibited distinct frequency patterns from wakefulness to NREM sleep[12]. Furthermore, the progression from wakefulness to NREM sleep was accompanied by the gradual breakdown of inter-regional fMRI based functional connectivity[13] and arousal dependent Hidden Markov Model (HMM) states[14]. In addition, the brain state dynamics, as broadly defined, has been widely investigated and implicated in cognitive processes and brain disorders[15–18].

More broadly, fMRI based resting-state functional connectivity has been increasingly recognized to be influenced by arousal fluctuations[19], as human subjects often exhibit notable arousal fluctuations or even sleep during resting-state fMRI. However, the BOLD signal is a hemodynamic signal and thus is also influenced by non-neural physiological factors (e.g., respiratory and cardiac signals) that may co-vary with arousal state changes[20]. Limited by the available neural recording and manipulation tools in humans, an animal sleep fMRI method is much needed as a platform to thoroughly disentangle arousal related neural and non-neural contributions to BOLD signals.

Clearly the macroscopic observations using non-invasive imaging techniques in humans and microscopic, circuitry level knowledge in animals are difficult to integrate, due to the vast gaps of spatiotemporal scales and species. Simultaneous electrophysiology and fMRI could potentially bridge the gap. However, most simultaneous electrophysiology and fMRI sleep studies were scalp EEG-fMRI in humans[21], which provided limited electrophysiological information due to the intrinsic limitation of scalp EEG. Meanwhile, invasive simultaneous electrophysiology and fMRI have been developed mostly in anesthetized animals[22] and not yet utilized in sleep research so far. Therefore, a sleep fMRI method based on simultaneously acquired invasive electrophysiology in mice could provide both local and global information and bridge the spatiotemporal and species gaps. Two major technical obstacles need to be overcome to achieve mouse sleep fMRI. First, it is difficult to perform un-anesthetized mouse fMRI because of the high stress level, due to the head restraining and loud noise of the fMRI environment, led alone to make mouse fall asleep. Second, it is well known that simultaneously electrophysiology and fMRI is highly challenging due to the mutual electromagnetic interference between electrophysiology and fMRI, which is more so in un-anesthetized animals.

To this end, we developed a mouse sleep fMRI method based on simultaneously electrophysiological recording at 9.4 T. To achieve this goal, we first established a highly MR-compatible electrophysiological recording setup in un-anesthetized mice at 9.4 T. Through further optimization on awake mouse fMRI, we achieved the recording of whole awake-sleep cycle from awake to NREM and REM states during fMRI. Importantly, using MR-compatible graphene fiber (GF) electrodes, hippocampal local field potential (LFP) signal were recorded and characteristic events such as spindle and SWR were extracted during sleep fMRI. With this mouse sleep fMRI method, we revealed global patterns of NREM and REM sleep, and more importantly, the global, asymmetric and sequential dynamics of state transitions, which was also evident in their trajectories in the low-dimensional manifold. Furthermore, utilizing long short-term memory (LSTM) recurrent neural network (RNN) modeling, we found that BOLD signals could predict state transitions, up to 17.8 s, prior to electrophysiological defined transition time point. Using the neural-event-triggered (NET) fMRI approach, we found SWRs had a

significantly higher BOLD responses in NREM state than in AW state, which could attribute to the co-occurrence of spindle events. In conclusion, this mouse sleep fMRI method will further advance mouse sleep research by providing both local and global view. Furthermore, it provides an accessible platform for elucidating the neural basis of arousal related BOLD signals. Combined with rich resources and tools in mice, the current method will help to establish a general multiscale framework of sleep.

## Results

### Sleep fMRI using MR-compatible electrophysiology recording in un-anesthetized mice

To investigate how arousal fluctuation contributes to the global dynamics, we established the mouse sleep fMRI setup using the simultaneous electrophysiology and fMRI. MR-compatible electrocorticography (ECoG) and depth electrodes were custom designed and fabricated for high MR-compatibility at 9.4 T, while maintaining good electrophysiological recording quality (Fig. 1a, b). Double-sided flexible printed circuit (FPC) with polyimide film as base substrate was used to minimize the thickness and width, and to maximize the flexibility of the array. Copper layer was used for its similar magnetic susceptibility to the brain (thus minimal MRI artefact), and gold was plated over copper layer in each recording site to avoid biological toxicity of copper. The 16-Channels ECoG electrodes covered a large portion of the cerebral cortex, including retrosplenial area (RSP), motor area, somatosensory area and posterior parietal association area (Fig. 1a and Supplementary Data 1). To record subcortical activity, we further fabricated a graphene fiber based depth electrodes. Graphene fiber was shown to be highly MR-compatible in our previous deep brain stimulation (DBS) study[23] at 9.4 T, and was repurposed for LFP recording in the hippocampus (Fig. 1b). The pipeline of electrode implantation surgery, habituation training and data acquisition was shown in Fig. 1c and described in Method. Both electrodes produced minimal MRI artifacts in T2 weighted anatomical images and T2* weighted functional images (Fig. 1d, e) of the relatively small sized mouse brain at 9.4 T.

With highly MR-compatible electrodes providing the feasibility of monitoring animal's arousal states, it remained challenging to record mouse sleep in the MRI environment. Based on our extensive previous experiences on awake mouse fMRI[24,25] and systematical optimization of stress level reduction[26], the current un-anesthetized mouse fMRI setup allowed simultaneous electrophysiological recording while minimizing animal's stress and thus facilitating sleep (Fig. 1f). In particular, the mouse's head was tilted for 30 degree and its forelimbs were allowed to move freely, both specifically designed for minimizing stress level[26].

It is well known that the simultaneously acquired electrophysiological signal suffers from severe MRI artifacts. Thus, we established and evaluated an off-line de-noising preprocessing pipeline (Supplementary Fig. 1), which greatly suppressed MRI artifacts to the extent that they no longer affected further analysis (Fig. 2a, b). Meanwhile, we also developed an fMRI preprocessing pipeline that was optimized for 4-hour sleep fMRI data. Raw fMRI data exhibited good (temporal) signal-noise-ratio (Fig. 2c) with small head motion and the "6 rp + 6 Δrp + 40 PCs" regression approach was modified from our previous studies[25,27] to minimize the effects of scanner drift, motion and other non-neural physiological noises (Fig. 2d and Supplementary Fig. 2). The 40 PCs were derived from fMRI signals outside the mouse brain (Supplementary Fig. 2c, d), largely capturing the non-neuronal signals[28], such as head motion (Supplementary Fig. 2e), physiological effects (Supplementary Fig. 2f–h) and infra-slow drift (Supplementary Fig. 2f, and i, j). Thus, using the "6 rp + 6 Δrp + 40 PCs" regression approach, arousal induced non-neuronal nuisance effects on fMRI signals were minimized. With this simultaneous electrophysiology and fMRI recording setup, a representative session of mouse sleep fMRI

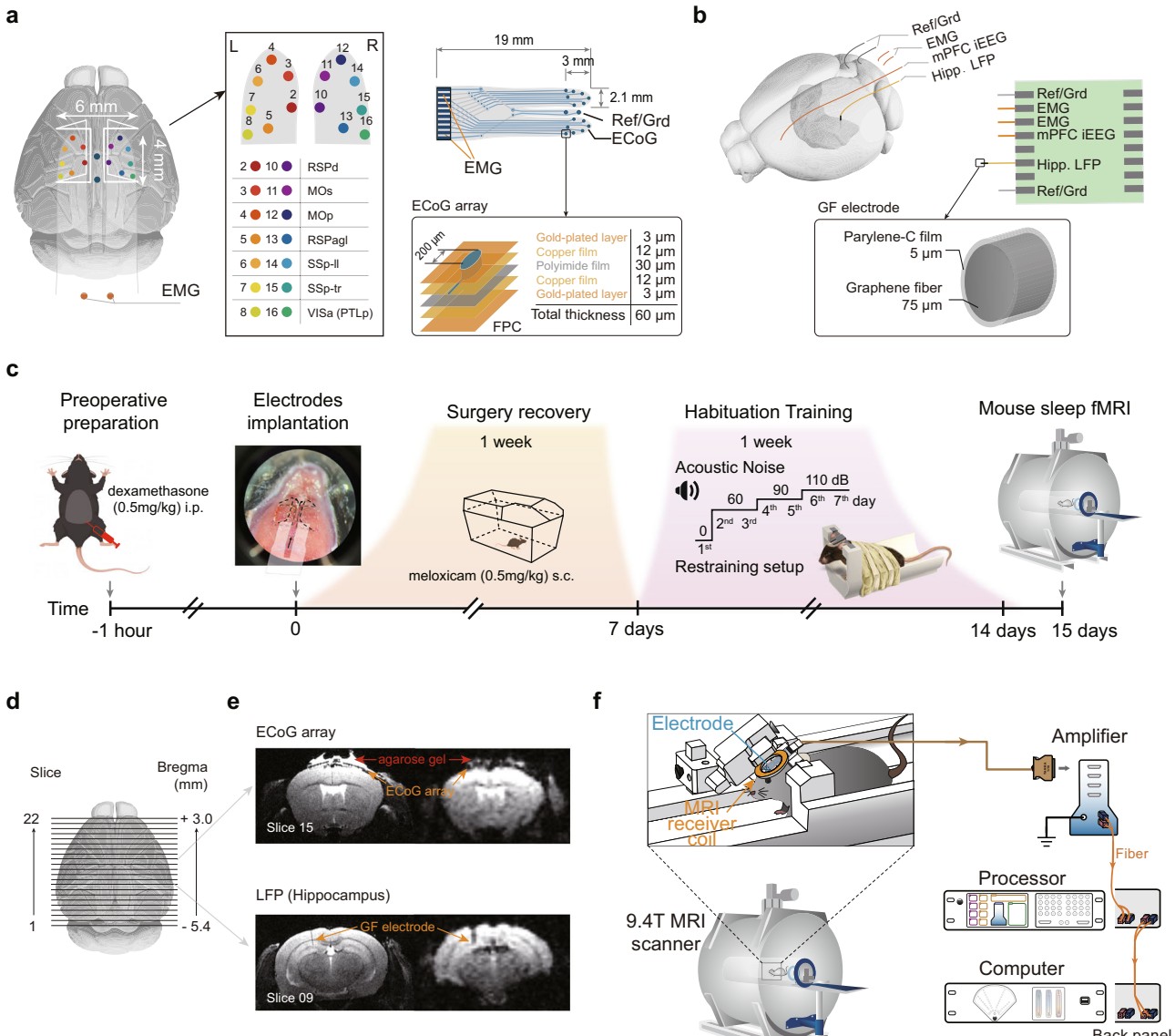

**Fig. 1 | Mouse sleep fMRI using MR-compatible electrophysiology recording in un-anesthetized mice. a** Design and location of MR-compatible ECoG array. The array was placed above the dura and centered on bregma (length, ~4 mm; width, ~6 mm) for electrocorticogram (ECoG) recording with two gold wires (orange) inserted in nuchal muscles for electromyography (EMG) recording. Right panel, the diagram of multi-electrode ECoG array and its layered construction. FPC, flexible printed circuit. **b** Design and location of depth electrode. Graphene fiber (GF) based MR-compatible depth electrode was implanted into the dorsal CA1 region of the left hippocampus with gold wires (orange) placed on dura and nuchal muscles for medial prefrontal cortex (mPFC) intracranial electroencephalography (iEEG) and EMG recordings, respectively. **c** The pipeline of animal surgery, recovery, habituation training and mouse sleep fMRI data acquisition with details described in Method. The cartoon mouse image was drawn by Figdraw. **d** Illustration of field of view (FOV) during simultaneously electrophysiological and fMRI recording. **e** Minimal MRI artifacts of ECoG arrays (upper) or GF depth electrodes (lower) on T2 weighted structural (left panel) and T2* weighted functional images (right panel) of the mouse brain. **f** Schematic illustration of the mouse sleep fMRI setup with details described in Method.

with awake (AW), NREM and REM sleep states was shown in Fig. 2e with denoised ECoG, electromyography (EMG) and fMRI signals. Sleep states were conventionally classified using electrophysiological signals (see Methods for details, Supplementary Fig. 3). Similar to previous optical imaging studies of mouse sleep[29], we utilized a 4-hour long fMRI acquisition scheme. Mice exhibited notable and increasing portion of NREM sleep as the scan progressed, and REM sleep was also observed during the 4-hour scan (Fig. 2f). In total, 46 4-hour sessions (184 h) simultaneous ECoG/LFP-fMRI recordings from 24 mice (27 ECoG-fMRI sessions from 14 mice and 19 LFP-fMRI sessions from 10 mice) were acquired in our dataset, including 3588 min of NREM sleep and 342 min of REM sleep (Fig. 2g). The above dataset with both raw and preprocessed data is openly accessible at https://doi.org/10.12412/BSDC.1668502646.20001.

## State dependent whole brain BOLD patterns and their neurophysiological correlates

Human sleep is characterized by long state durations, e.g., each NREM-REM cycle is approximately 90 min[30], which prevents conventional general linear model (GLM) analysis due to its low frequency beyond the detection limit of fMRI. However, mouse sleep is fragmented with short state durations[31] (Supplementary Fig. 4, median duration of AW: 46.0 s, NREM: 41.0 s and REM: 121.5 s in our data), thus is convenient for analyzing state dependent activations. Using the conventional GLM analysis, we mapped the brain-wide BOLD activations of NREM and REM sleep, relative to the AW state (Fig. 3). For the NREM state, a large part of cerebral cortex and hippocampus were activated, while subcortical regions, such as thalamus and part of midbrain, were deactivated compared to the AW state (Fig. 3a–c).

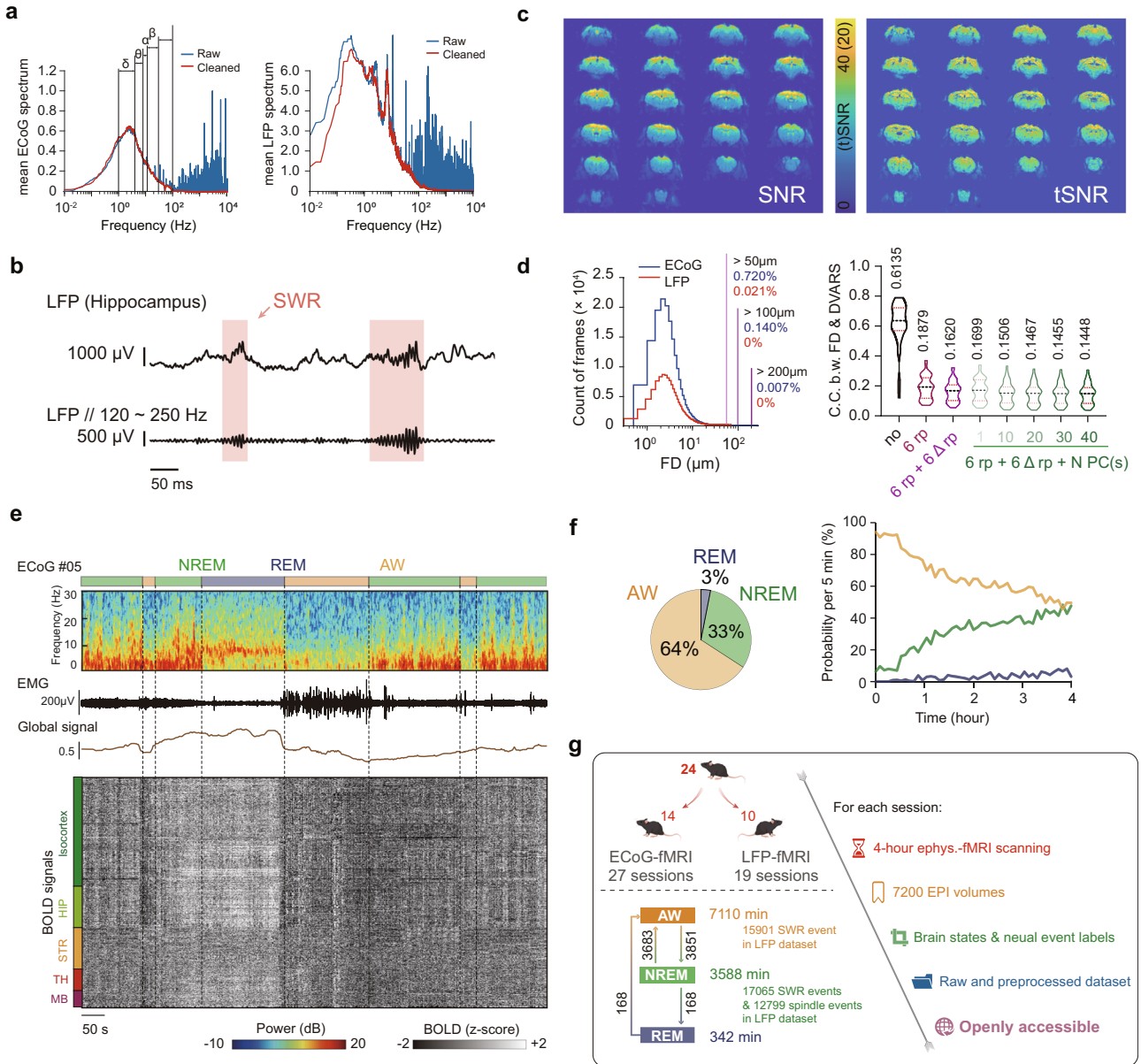

**Fig. 2 | Overview of mouse sleep fMRI dataset. a** Mean power spectrum of raw and denoised ECoG signals for ECoG-fMRI sessions and LFP signals for LFP-fMRI sessions. **b** Representative denoised LFP and EMG signals during fMRI scanning. Upper panel, broadband (0.5-512 Hz) LFP signal; Lower panel, band-pass filtered (120-250 Hz) LFP signal. Red shadow indicated sharp wave ripple (SWR) events. **c** Representative example of (temporal) signal-to-noise, i.e., (t)SNR, of our raw fMRI signal. **d** Small head motion level of ECoG/LFP-fMRI sessions during fMRI scanning (left panel) and minimal head motion effects on BOLD signals after the "6 rp + 6 Δrp + 40 PCs" regression (right panel). Notably, the in-plane spatial resolution of EPI images was 200 μm. The number above each violin plot represented the mean Pearson's correlation coefficients (C.C.) between frame-wise displacement (FD) and DVARS. **e** A representative session of mouse sleep fMRI with AW, NREM and REM sleep states (Mouse 26, ch5, 7400s–8400s). From upper to lower panels: power spectrogram of ECoG signal, amplitude of EMG signal, global BOLD signal and voxel-wise whole-brain BOLD signals. AW, awake; NREM, non-rapid eye movement; REM, rapid eye movement. **f** Averaged (left panel) and time dependent (right panel) probability of each brain state during sleep fMRI scanning (*n* = 46 sessions from 24 mice). **g** Overview of mouse sleep fMRI dataset. The number of events here were counted under the sampling rate of 1024 Hz. Cartoon mouse images were drawn by Figdraw. The dataset is openly accessible at https://doi.org/10.12412/BSDC.1668502646.20001. Source data are provided as a Source Data file.

Importantly, using the simultaneously acquired ECoG or LFP signals, we explored the potential relationship between sleep state dependent BOLD and electrophysiological signals. We calculated hemodynamic response function (HRF)-convolved ECoG/LFP band-limited power and its correlation with BOLD changes (Supplementary Fig. 5). The relative changes of ECoG or hippocampal LFP were in generally consistent with current understanding of NREM and REM sleep[29,31] (Fig. 3d, e, left panels). Importantly, we found a significant broadband (1–100 Hz) correlation between the relative ECoG power and relative cortical BOLD signal changes (Fig. 3d, right panel) during

NREM state, which was also evident in hippocampal CA1 recordings (Fig. 3e, right panel). For REM state, significant correlation between theta band-limited power and BOLD changes was observed (Fig. 3d, e, right panels), suggesting the potential relationship of theta band and BOLD changes.

**Low dimensional dynamics within and between brain states**

The above NREM and REM activation patterns were state-dependent features. However, previous studies indicated that sleep architectures are dynamic[2]. Taking advantages of whole-brain mouse

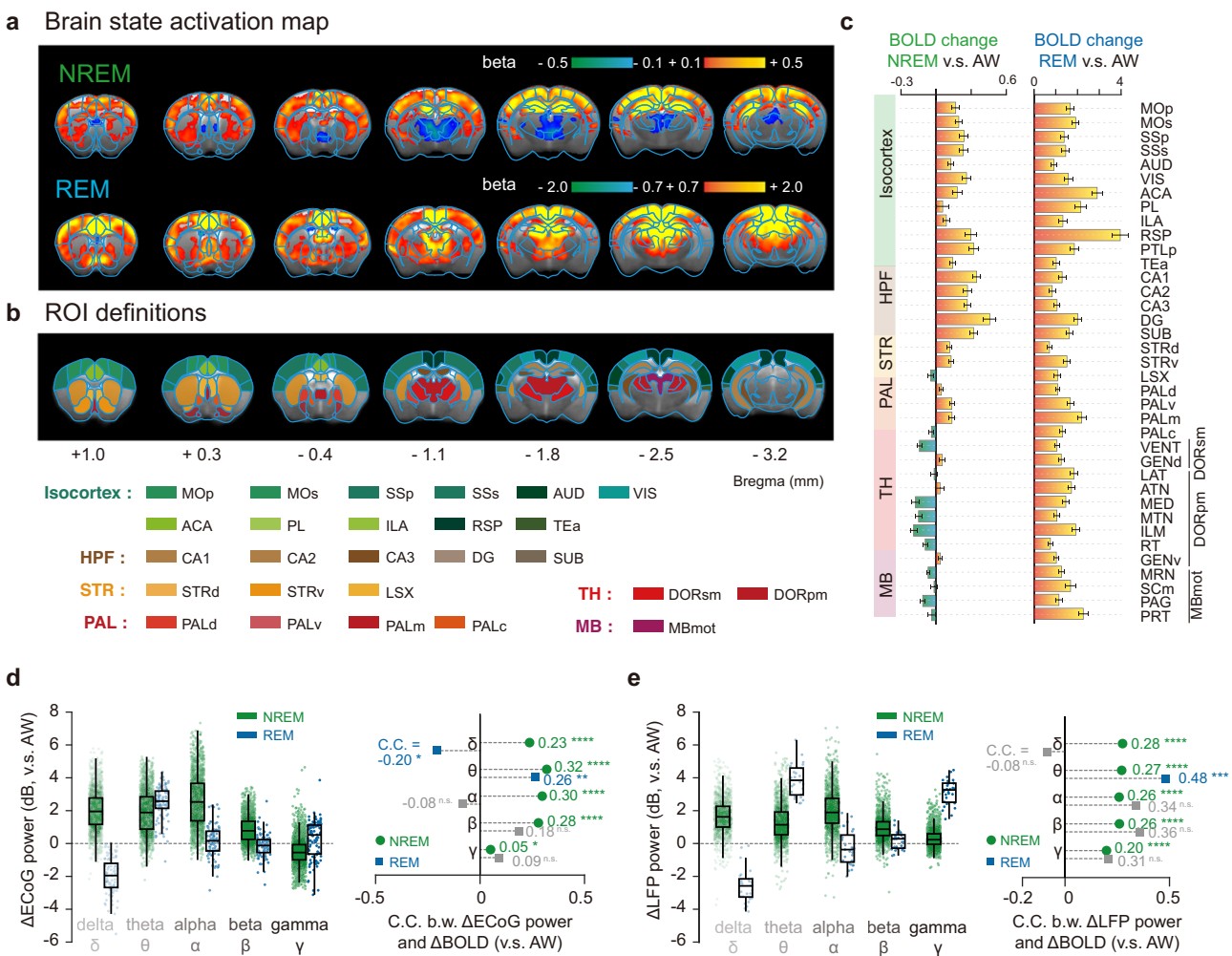

**Fig. 3 | Brain-wide BOLD activations of NREM and REM sleep and their electrophysiological correlates. a** Group BOLD activation maps of NREM and REM compared to AW state (FDR corrected, $p < 0.05$; $n = 46$ sessions from 24 mice). **b** Region-of-interest (ROI) definitions for extracting the BOLD changes in (**c**). Abbreviations of ROIs were listed in Supplementary Data 1. **c** Mean (+/− SEM) relative BOLD changes of NREM and REM compared to AW state (sample size: $n = 46$ sessions from 24 mice). **d**, **e** Electrophysiological correlates of sleep dependent BOLD changes. Left panel, relative ECoG (**d**) or LFP (**e**) band-limited power in NREM and REM compared to AW state. Right panel, Pearson's correlation coefficients (C.C.) between relative ECoG (**d**) or LFP (**e**) band-limited power and ΔBOLD (v.s. AW)

BOLD response in NREM and REM states. delta (δ), 1–4 Hz; theta (θ), 5–10 Hz; alpha (α), 11–20 Hz; beta (β), 21–40 Hz; gamma (γ), 41–100 Hz. Statistical significance was calculated by two-tailed $t$-test.*$p < 0.05$; **$p < 0.01$; ****$p < 0.0001$; n.s., no significance. For ECoG dataset (**d**), the sample size of NREM is 2372 and REM is 138 (sample size: $n = 27$ sessions from 14 mice). For LFP dataset (**e**), the sample size of NREM is 1479 and REM is 30 (sample size: $n = 19$ sessions from 10 mice). The box showed the first and third quartiles; inner line was the median over sessions; whiskers represented minimum and maximum values (outliers removed). Source data are provided as a Source Data file.

sleep fMRI data, we investigated the low dimensional dynamics within and between brain states. First, we conducted group principal component analysis (PCA) to BOLD signals for dimensional reduction. The −100th PC accounted for at least 0.3% explained variances and accumulatively the first 100 PCs resolved greater than 78% explained variances of BOLD signals. (Fig. 4a, Supplementary Fig. 6 and Supplementary Data 2).

The first four PCs were shown in Fig. 4a, b. Each PC exhibited non-stationary temporal weights from the start to the end of each state (Fig. 4c), suggesting dynamic involvements of different functional networks within brain states. Further dynamics were expected across brain states, thus we characterized the state transition probabilities (Fig. 4d). Electrophysiological and whole brain BOLD features were shown in Fig. 4e (upper panel) and Supplementary Fig. 7–13 for four state transitions, respectively. For each state transition, we observed sequential BOLD signal fluctuations traversing the mouse brain (Supplementary Fig. 7 and Supplementary Data 3). Further global spatiotemporal patterns of all four state transitions were shown in

Supplementary Fig. 8–11. Importantly, the temporal weights of PCs (tPCs) also exhibited diverse characteristics across state transitions (Fig. 4e, lower panel, and Supplementary Fig. 13), suggesting dynamic involvements of different functional networks across brain states. Moreover, in the low dimensional spaces spanned by BOLD PCs (Fig. 4f and Supplementary Fig. 14) and electrophysiological band-limited power ratios (Fig. 4g), activity flows of the manifolds exhibited dynamic properties within and across brain states. Interestingly, we found the separated trajectories between "AW to NREM" and "NREM to AW" transitions (Fig. 4f, g). And we further quantified such phenomenon and the significant asymmetry were observed around state transitions in both BOLD and electrophysiological space (Fig. 4h), suggesting "AW to NREM" and "NREM to AW" transitions were asymmetric processes.

**Predictions of state transitions using the LSTM RNN model**
The dynamic characteristics of PCs across state transitions promoted us to investigate whether such transitions could be predicted by BOLD

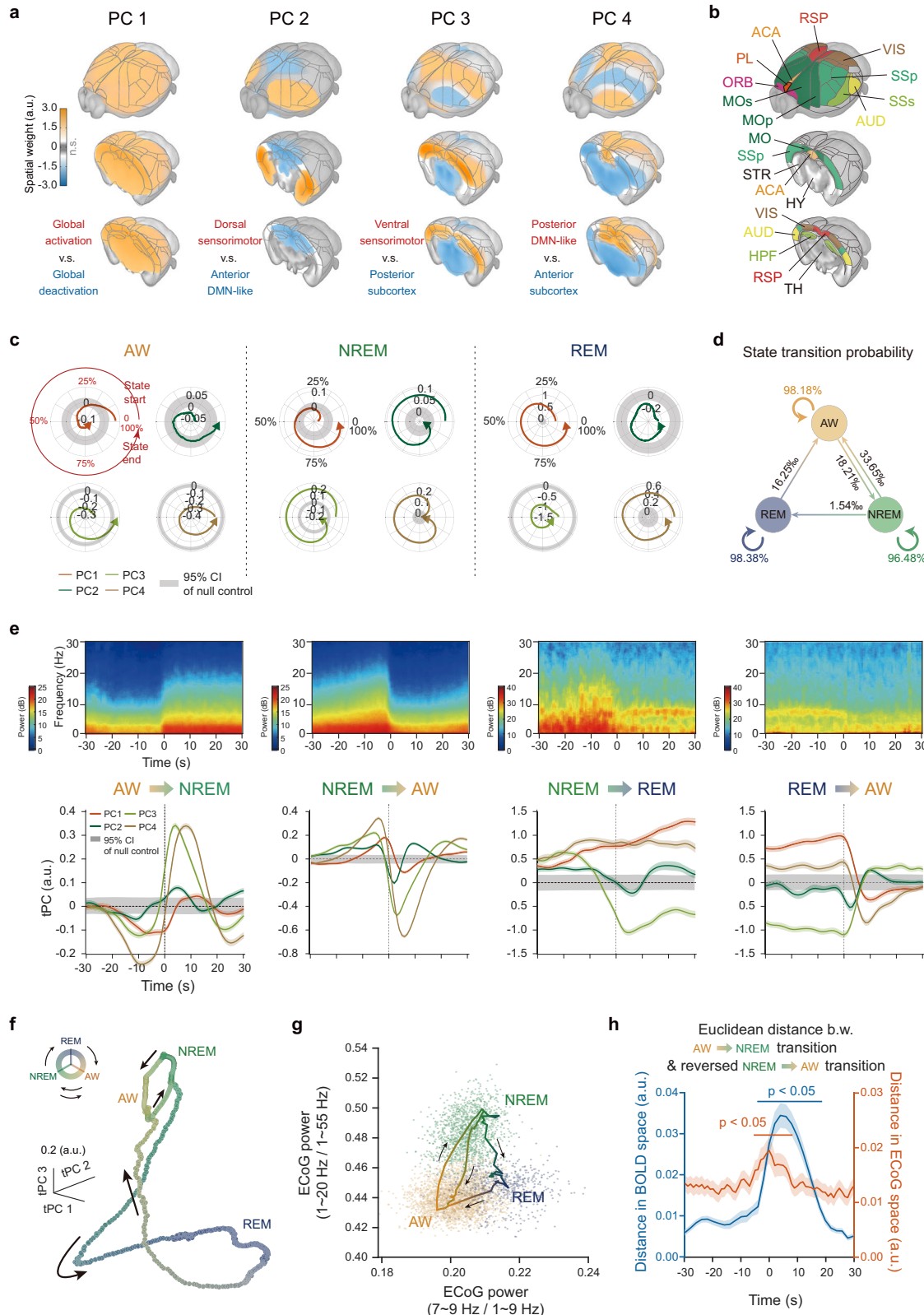

signals prior to electrophysiology defined transition time, and if so, which functional networks contributed to such predictions. Thus, we established a LSTM RNN model for state transition prediction, with BOLD signals (tPCs) preceding state transitions as model input and the brain state after such transition as model output (Fig. 5a). We systematically evaluated the model parameters, and the resulting optimized parameter set was 1 hidden LSTM layer with 50 hidden units

(Fig. 5b). Prediction accuracy on the validation dataset was primarily related to the gap times preceding state transitions (Fig. 5c). Using the optimized parameter set, we found high prediction accuracy (Fig. 5d) of more than 85% across all seven prediction categories.

The high prediction accuracy gradually decreased as the gap time increased (Fig. 5e–h, upper panel), and the mean discriminate times were 10.55 s for "AW to NREM", 3.28 s for "NREM to AW", 17.84 s for

**Fig. 4 | Low dimensional dynamic signature across brain states. a** Spatial maps for the first four principal components (PCs) of BOLD signals. **b** Major brain divisions on 3D surface viewing as in (**a**). Abbreviations of ROI names were listed in Supplementary Data 1. **c** Circular distribution of temporal weights of PCs (tPCs) from the start to the end of each brain state: AW (left, $n = 2703$ epochs), NREM (middle, $n = 2498$ epochs) and REM (right, $n = 165$ epochs). Epochs with short durations (<60 s) were excluded and further applied to the following analysis. Radius of radar plots, tPCs; Color line, mean tPCs; gray shadow, the 95% confidence interval (CI) of null control (1000 shuffling). **d** Transition probability of brain states. **e** Averaged electrophysiological power spectrogram (upper panel) and mean (+/− SEM) tPCs of BOLD signals (lower panel) relative to brain state transitions ("AW to NREM", $n = 1803$ epochs; "NREM to AW", $n = 1720$ epochs; "NREM to REM", $n = 140$ epochs; "REM to AW", $n = 128$ epochs). Gray shadow, the 95% CI of null control. **f** Low dimensional manifold of BOLD signals traversed by the brain state across first three PCs, with arrows depicting the directions of flow along the manifold. **g** Similar to (**f**) but in two-dimensional electrophysiological space. Color dots, states per second. **h** Asymmetric trajectories of brain state transitions between AW and NREM states. Large distance of low dimensional manifolds in electrophysiological (**f**) and BOLD (**g**) spaces between "AW to NREM" and reversed "NREM to AW" transitions. Statistical significance was calculated by one-tailed $t$-test. Colored shadow, SEM. a.u. arbitrary unit. Source data are provided as a Source Data file.

"NREM to REM", and 3.45 s for "REM to AW" state transitions (Fig. 5e–h, middle panel). The null dataset was constructed by shuffling prediction categories. Correspondingly, sensitivity results of the LSTM RNN model revealed several regions significantly contributed to the prediction of state transitions (Fig. 5e–h, lower panel, and Supplementary Fig. 15), including (1) ventral thalamus (vThal), medial mammillary nucleus (MM), RSP, ventral tegmental area (VTA), median raphe nucleus (MnR), globus pallidus (external segment, GPe) and interposed cerebellar nucleus (Int) for "AW to NREM" state; (2) anterior cingulate area (ACA), agranular insular area (AI), visual areas (VIS), pontine reticular nucleus (ventral part, PnV), hippocampal formation (HPF) and primary somatosensory area (SSp) for "NREM to AW" state; (3) paragigantocellular reticular nucleus (PGRN), periaqueductal gray (PAG), HPF, VIS, medial preoptic area (MPO), supplemental somatosensory areas (SSs) and amygdala areas (AMY) for "NREM to REM" state; and (4) MM and pontine central gray (PCG) for "REM to AW" state.

Interestingly, we found HPF exhibited significant contributions on the prediction of state transitions from NREM to AW and REM states. Utilizing the GF electrode placed in hippocampus CA1 region of our LFP-fMRI dataset, we compared the LFP power spectrum between "NREM to AW" and "NREM only" states, and found significantly higher power in three frequency bands, including (1) 7-37 Hz from −14 to −2 s, (2) 44-95 Hz from −12 to −1 s, and (3) 119-205 Hz from −11 to −3 s before state transitions (Fig. 6a). Also, we conducted same analysis between "NREM to REM" and "NREM only" states, and found significantly higher power in two frequency bands, including (1) 1-34 Hz from (beyond) −30 to −3 s, and (2) 128-141 Hz from (beyond) −30 to −13 s before state transitions (Fig. 6a). These "NREM only" epochs were randomly selected from the sleep fMRI dataset, and epochs of "NREM to AW", "NREM to REM" and "NREM only" with short durations (<60 s) were excluded. Then, two sample t-test was conducted with threshold of $p < 0.05$. Furthermore, we isolated SWR and spindle events, among other events, based on LFP dataset (Fig. 6b). Altogether, SWR and spindle events showed a pronounced decrease of their occurrence probability before "NREM to AW (or REM)" state transition (Fig. 6c, d). To be note, the preceding time of SWR event changes (Fig. 6c) was similar to the mean discriminate time of "NREM to AW (or REM)" state transition (Fig. 5f, g, middle panel) upon LSTM RNN predictions, further suggesting the validity of the above LSTM RNN prediction.

### State dependent global spatiotemporal pattern of SWRs

Different arousal states are characterized by various neurophysiological events, such as abovementioned spindle and SWR in NREM sleep, sawtooth wave and pontine-geniculo-occipital wave in REM sleep. SWRs during AW or NREM state have been shown to play an important role in memory consolidation[6], but SWRs related global spatiotemporal pattern has not been fully explored, especially in different arousal states. Spindle and SWRs events were identified (Supplementary Fig. 16) based on previous studies[5,7]. Using the neural-event-triggered (NET) fMRI approach[22], we observed SWRs evoked BOLD activations in HPF, RSP and ventral sensorimotor cortex (vSMC) during NREM state (Fig. 7a, Supplementary Fig. 17–18 and Supplementary

Data 4). Interestingly, we also found significant deactivations in subcortical regions such as thalamus, hypothalamus and midbrain regions (Fig. 7a and Supplementary Fig. 18). In AW state, we obtained similar but much weaker spatiotemporal profiles (Fig. 7a, b and Supplementary Fig. 17), suggesting state dependency of SWRs evoked global patterns. Thus, we quantified above differences of SWRs evoked BOLD responses in medial prefrontal cortex (mPFC) and HPF (Fig. 7c and Supplementary Fig. 19) and found significantly higher BOLD responses in NREM state than those in AW state. Such difference of BOLD responses in mPFC was also significantly correlated with the difference of electrophysiological power in 3-45 Hz (Fig. 7d).

In agreement with previous studies[5], SWRs exhibited a strong co-occurrence with spindle events (10-16 Hz) in our data (Fig. 7e and Supplementary Fig. 20). Thus, we explored whether the state dependent SWRs evoked BOLD response could attribute to the co-occurrence of spindle events. Using the same NET fMRI approach, we revealed that the spatiotemporal pattern of spindle-uncoupled SWRs in NREM state (Fig. 7f, upper panel, Supplementary Fig. 21 and Supplementary Data 4) resembled the pattern of SWRs in AW state. And spindle-coupled SWRs exhibited similar spatiotemporal pattern but higher cortical activations, compared to the spindle-uncoupled SWRs (Fig. 7f, lower panel, and Supplementary Fig. 22). Difference of spindle-coupled and -uncoupled SWRs evoked BOLD responses (Fig. 7g) was significant and highly correlated with the difference of electrophysiological power in 4-42 Hz (Fig. 7h). Moreover, there was no significant difference of BOLD responses between SWRs in AW state and spindle-uncoupled SWRs in NREM state (Fig. 7i and Supplementary Fig. 19). The above results suggested that spindle co-occurrence enhanced the global activations of SWRs evoked BOLD responses, which was consistent with global positive BOLD responses of SWR-uncoupled spindle (Supplementary Fig. 23). Therefore, the above results suggested that the co-occurrence of spindle events contributed to the enhanced SWRs evoked BOLD responses in NREM state.

To investigate whether there were any synergistic effects of the SWRs and spindles triggered BOLD responses, we firstly summed the BOLD responses of solitary spindles (Fig. 8a and Supplementary Fig. 23–24) and SWRs in each session ("summed responses"), as well as those of spindle coupled SWRs ("coupled responses"). Then, we conducted paired t-test across sessions between the above coupled and summed responses at regional (Fig. 8b–d) and whole brain levels (Fig. 8e). We found a "coupled > summed" response across cortical regions and "coupled < summed" response in thalamus (Fig. 8b–e), suggesting synergistic effects of the SWRs and spindles triggered BOLD responses. Slow waves are thought to participate in the regulation of NREM sleep process. Using the same NET-fMRI approach, we also calculated the slow wave triggered spatiotemporal map and found the BOLD activations in RSP and thalamus (Supplementary Fig. 25).

## Discussion

In the current study we developed the mouse sleep fMRI method based on simultaneous electrophysiological and fMRI recording in un-

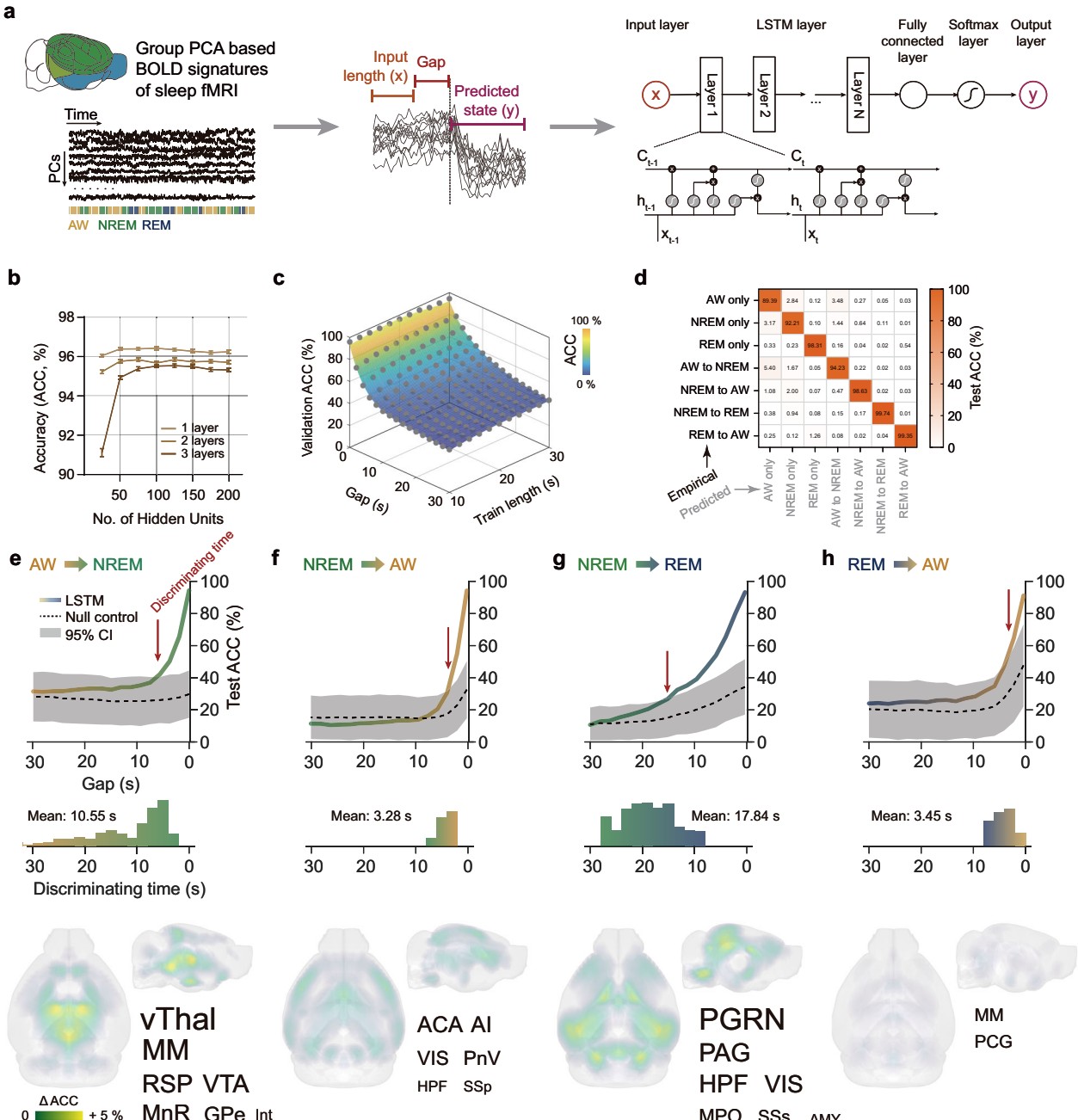

**Fig. 5 | LSTM RNN Prediction of state transitions based on large-scale BOLD signatures. a** Computational pipeline of the LSTM RNN model for state transition prediction. The model input was the BOLD signals (tPCs) preceding state transitions, and the output was the brain state after state transitions. Details were described in Methods part. **b** Mean (+/− SEM, $n = 500$ bootstrap sampling, same in all further analysis) prediction accuracy (ACC) of the LSTM RNN model on the validation dataset with different numbers of layers and hidden units. **c** Prediction accuracy on the validation dataset was primarily related to the gap times preceding state transitions. Results were based on the LSTM RNN model with 1 hidden layer and 50 hidden units. Gray dot and colored shade represented an individual result and the corresponding best quadratic fitting. **d** Confusion matrix between empirical and predicted seven categories showed high prediction accuracy on the testing dataset based on 10 s input length, 0 s gap times, 1 hidden layer and 50

hidden units (used in all further analysis, except the gap time). **e–h** Gap time dependent test accuracy and regional sensitivity on brain state predictions: AW to NREM (**e**), NREM to AW (**f**), NREM to REM (**g**) and REM to AW (**h**). Upper panel, high prediction accuracy of state transition preceding electrophysiology defined transition time. Note the null dataset was constructed by shuffling categories. Gray shadows: 95% confidence intervals on the null control dataset. Middle panel, distributions of discriminate times of brain state prediction compared to null dataset. Lower panel, sensitive regions of LSTM RNN model on brain state predictions (FDR corrected $p < 0.05$ and Cohen's $d > 0.3$). Results in the axial view were shown in Supplementary Fig. 15. Font sizes of sensitive regions were scaled according to the changes of prediction accuracy. Abbreviations of these regions were listed in Supplementary Data 1. Source data are provided as a Source Data file.

anesthetized mouse at 9.4 T. Our method provided the global view of sleep state-dependent patterns and state-transition dynamics. Furthermore, we revealed the potentiated SWR-evoked BOLD response in NREM state compared to that in AW state, largely attributed to

the co-occurrence of spindle events. Therefore, our method demonstrated the unique capability of revealing the global sleep dependent features, and provides an accessible platform for sleep research and investigation of neural basis of the arousal related fMRI signal.

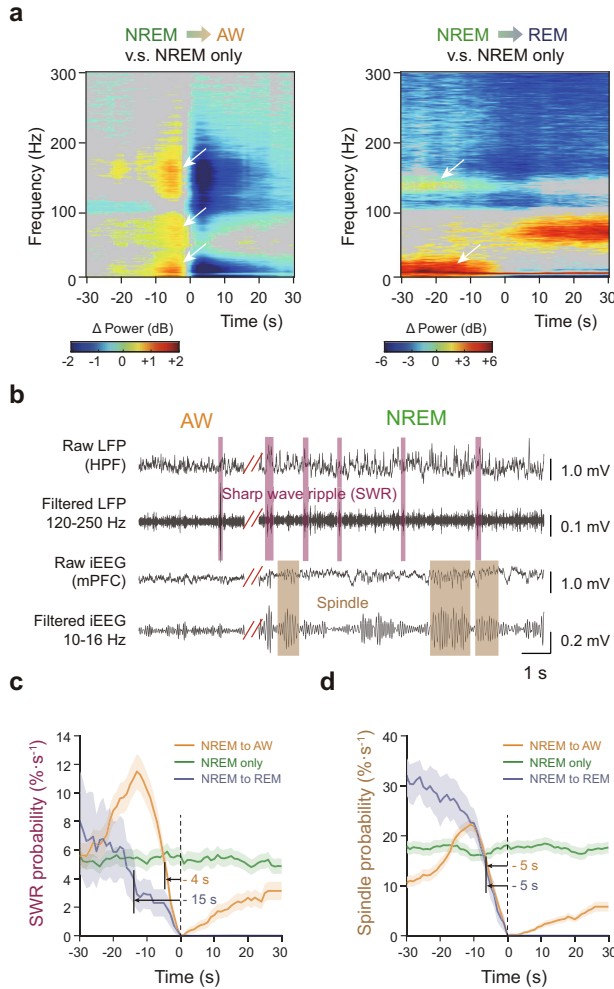

**Fig. 6 | Hippocampal involvement in state transition dynamics and prediction.** **a** Differences of power spectrograms between "NREM to AW" (720 epochs) or "NREM to REM" (29 epochs) transition and "NREM only" (2521 epochs) state (two-sample t-test, two tails, $p < 0.05$). **b** Representative trace of sharp wave ripples (SWRs) and spindle events. Broadband (raw, 0.5-512 Hz) LFP or iEEG signal together with their band-pass filtered (LFP: 120-250 Hz, iEEG: 10-16 Hz) signals. Purple and brown shadows represented SWRs and spindle events, respectively. **c, d** Significant decrease of SWRs (**c**) and spindle (**d**) probabilities per second prior to state transitions, compared to "NREM only" state (two-sample t-test, left tail, $p < 0.05$). Colored numbers, the earliest time of significantly lower event probabilities. Color lines (or shadows), mean (+/− SEM.) event probabilities. Source data are provided as a Source Data file.

It is well known that simultaneous recording of electrophysiological and fMRI signals is technically challenging, particularly in un-anesthetized animals. Such difficulty mainly arises from the mutual electromagnetic interference and head motion related complications in awake animals. To achieve high MR-compatibility, our 16-Channel ECoG electrodes and GF depth electrodes were specially designed to achieve minimal MRI distortions and high-quality electrophysiological recordings. Such good compatibility is important for simultaneous electrophysiological fMRI recording in mice, as the mouse brain is much smaller than brains of human and monkey, thus it is more prone to imaging artifacts of electrodes. With these developed electrodes, after de-noising procedures (Fig. 2a, Supplementary Fig. 1 and Supplementary Data 5), the characteristics of electrophysiological signals were in line with those outside the MRI environment, such as amplitude and power spectrum[29,31].

Interestingly, we did not observe typical pulse artifact from the ECoG and LFP power spectrums (Fig. 2e and Fig. 4e), which is often problematic in human EEG-fMRI studies[32]. After our electrophysiological denoising procedure, no notable ballistocardiogram artifact was observed in 5–10 Hz frequency band, corresponding to the unanesthetized mouse heart rate between 300 and 600 beats per minute. This phenomenon might attribute to our intracranial placement of electrodes and the stable electrode fixation. In human EEG-fMRI studies, pulse artifact occurs when an EEG electrode is placed over a pulsating vessel[32]. The pulsation can cause slow electrode movements that contaminate EEG activities. Our intracranial electrodes were tightly fixed on the mouse skull by dental cement. Thus, scalp pulse and cardiac-related motion were less likely to impact the electrode movement and further influence our electrophysiological signals.

However, previous studies combined electrophysiology and fMRI were mainly conducted in anesthetized animals, e.g., rat and macaque, which prevented the utilization in sleep research. Based on our extensive experience on awake mouse fMRI[24,25], in the current study we further demonstrated that mouse could sleep in the noisy MRI environment. This critical improvement paved way for multimodal fMRI research in mice, and would also be highly meaningful for other sleep related MRI investigation such as diffusion MRI[33].

Mouse fMRI is uniquely suited to reveal the state dependent global patterns and state transition dynamics, as its sleep is fragmented with short state duration and thus frequent state transitions[31]. In contrast, the long sleep state durations of human and monkey prevent the conventional GLM analysis to map the state dependent activations, due to its low frequency beyond the detection limit of fMRI. For example, each NREM-REM cycle is approximately 90 min in humans[30]. Such long cycles also prevent fMRI analysis of state transitions as there would be so few transitions in each scan. For comparison, our 184 h mouse sleep fMRI data included 3851 "AW to NREM", 3683 "NREM to AW", 168 "NREM to REM" and 168 "REM to AW" transitions (Fig. 2g), which enabled further detailed analysis of state transitions. The dynamics of broadly defined state transition and fluctuations in resting-state fMRI is now widely investigated, and has been implicated in cognitive processes[15,16] and brain disorders, e.g. Alzheimer's disease (AD)[17] and obsessive–compulsive disorder (OCD)[18]. The functional network of those patients with brain disorders exhibited abnormal dynamic rhythms[17,18], indicating potential clinical relevance. And also, consciousness has been shown to modulate the diversity of the state dynamics across different sedation levels[34]. Utilizing the rich transgenic mouse disease models, e.g., various AD mouse models, future research can be conducted based on our mouse sleep fMRI setup to further investigate the mechanisms of state transition dynamics and its role in brain disorders.

The abundant sleep transitions in mouse sleep fMRI greatly facilitated the investigation of the macroscopic cerebral dynamics, for which we utilized group PCA. A major advantage of PCA over other analytic approaches is that it imposes orthogonality onto the components, which is crucial for providing a low-dimensional subspace to embed the state space manifold. Other dimension reduction methods (such as independent component analysis) find a different set of optimal solutions (such as maximal statistical independence), but these are not, in general, linearly independent. Notably, state space attractors (Fig. 4f, g) are invariant to linear transformations of their embedding phase space as long as the dimensions remain orthogonal. Hence, PCA enables analysis of the state space trajectory (or flow), which reflects the temporal evolution of the global brain state[35]. Combined with rich resources and tools in mice, more features of dynamic microstructure within mouse sleep states can be explored in future studies. Particularly, the asymmetrical trajectories in both electrophysiological and BOLD spaces indicated the asymmetry between "AW to NREM" and "NREM to AW" transitions (Fig. 4h), further suggesting different neural circuits for awake-promoting and NREM sleep-promoting processes. Our results provided a framework for

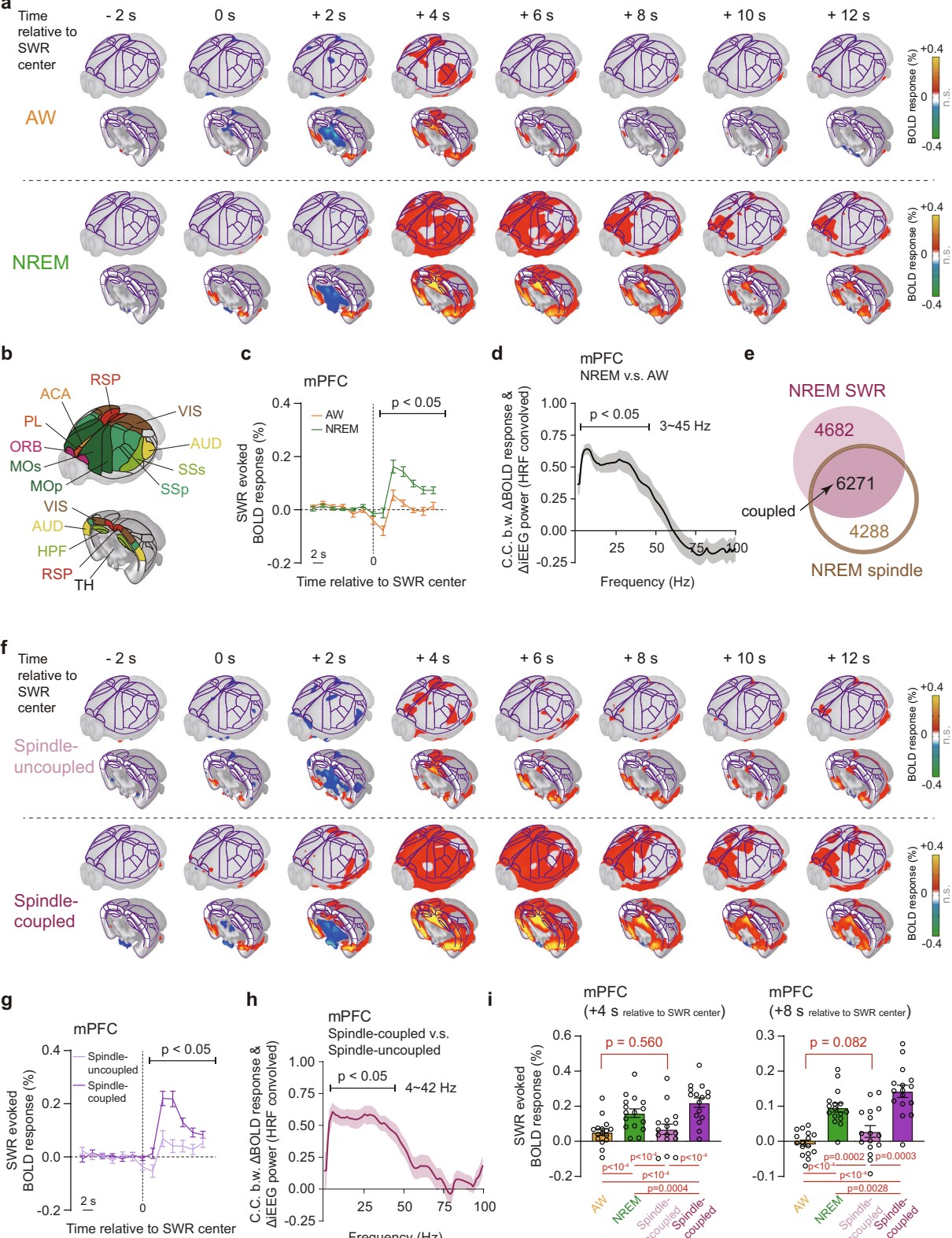

sleep research through the lens of complex dynamical systems, linking the flow of electrophysiological signatures to the dynamic reconfiguration of functional networks in the low-dimensional state space.

The within- and between-state dynamics was also clearly demonstrated by prediction of state transitions prior to electrophysiology defined transition time point using LSTM RNN models[36] on BOLD signals. The long preceding discriminate times (AW to NREM: 10.55 s,

and NREM to REM: 17.84 s) were significantly longer than previous reported neurophysiological results and considering the hemodynamic delay of 1.5–2 s in mice[25], it may be slightly longer if direct neural signals were used. For example, the firing rate of sleep-promoting neurons in the POA and wake-promoting neurons in the Locus Coeruleus (LC), tuberomammillary nucleus (TMN), and BF altered before the transition for less than 1 s[37], and the firing rate of GABAergic neuron

**Fig. 7 | State dependent global spatiotemporal patterns of SWRs in AW and NREM states using neural-event-triggered (NET) fMRI. a** SWR-triggered BOLD responses in AW (12732 epochs) and NREM (10953 epochs) states. The number of event epochs were counted under the sampling rate of 0.5 Hz (fMRI repetition time) in the following analysis. Results in the axial view were shown in Supplementary Fig. 17, 18. **b** Major brain divisions on 3D surface viewing as in (**a**). Abbreviations of ROIs were listed in Supplementary Data 1. **c, d** Significant difference of SWR-triggered BOLD responses in mPFC between AW and NREM states (**c**) and its corresponding electrophysiological correlates (**d**). Gray lines (or shadows), mean (+/− SEM.) correlation. Sample size: *n* = 16 sessions. **e** Overlap of SWRs and spindle events in NREM state. **f** Spindle-uncoupled (4682 epochs) and coupled

(6271 epochs) SWR-triggered BOLD responses in NREM state. Results in the axial view were shown in Supplementary Fig. 21, 22. **g, h** Significant difference of BOLD responses in mPFC between spindle -uncoupled and -coupled SWRs in NREM state (**g**) and its corresponding electrophysiological correlates (**h**). Colored lines (or shadows), mean (+/− SEM.) correlation. Sample size: *n* = 16 sessions. **i** No significant difference of event evoked BOLD signals in mPFC between awake SWRs and spindle-uncoupled SWRs. Each dot represented an individual session. mPFC, medial prefrontal cortex. Error bars, standard errors of the mean. Statistical significance was calculated by two-tailed *t*-test. Source data are provided as a Source Data file.

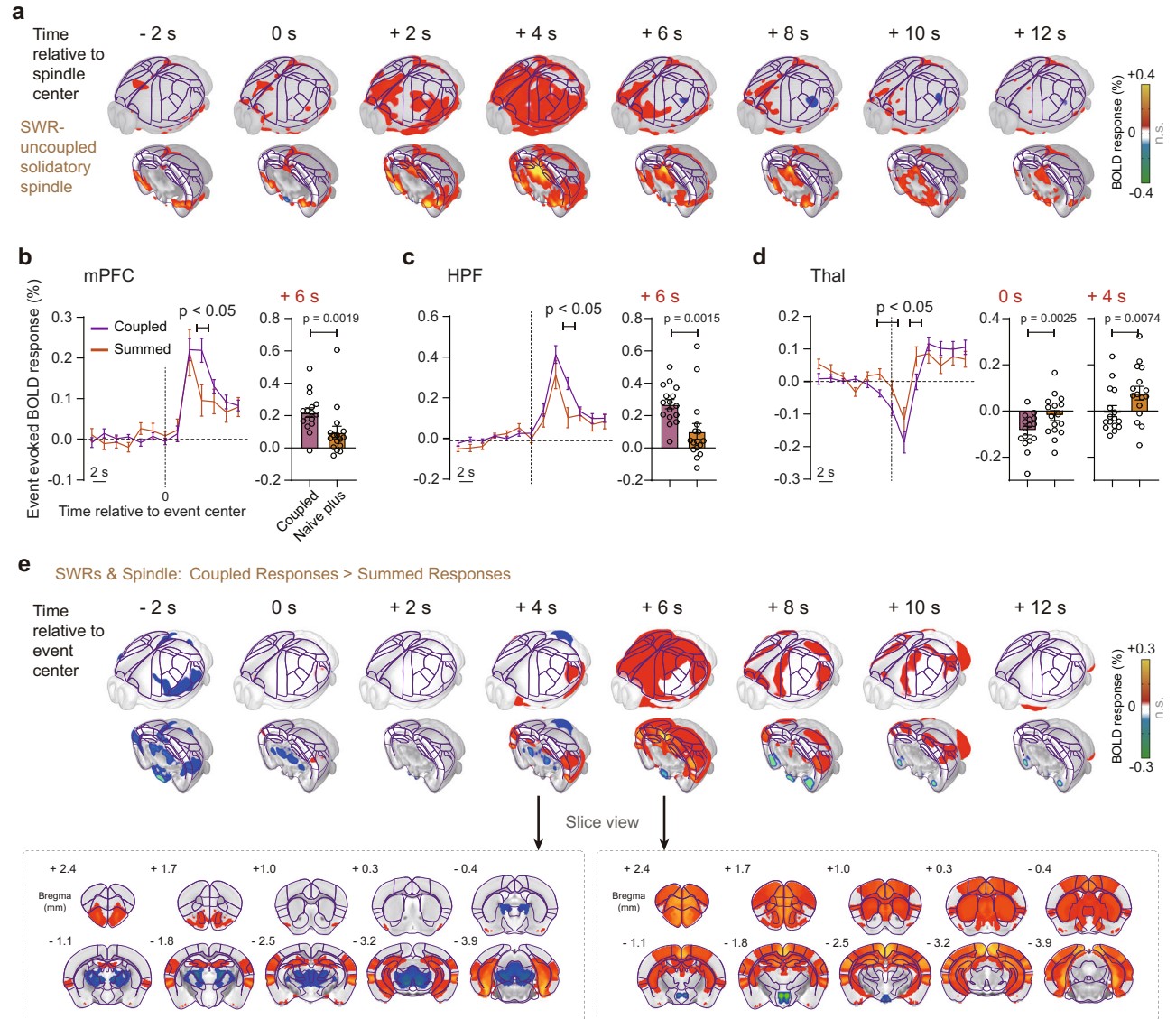

**Fig. 8 | Synergistic effects of SWR-spindle coupling. a** Solitary spindle (*n* = 4288 epochs) triggered BOLD responses in the NREM state. Results in the axial view were shown in Supplementary Fig. 23. **b–d** Significant differences between spindle coupled SWRs triggered BOLD response and the summation of BOLD responses of solitary spindles and SWRs in mPFC, HPF and Thal. Each dot represented an individual session (sample size: *n* = 16 sessions). Statistical significance was tested using

the two-tailed paired *t*-test. Error bars, standard errors of the mean. **e** Whole brain mapping of the synergistic effects of SWR-spindle coupling. Statistical significance was tested using the two-tailed paired *t*-test. mPFC medial prefrontal cortex, HPF hippocampal formation, Thal thalamus. Source data are provided as a Source Data file.

in ventral medulla (vM) increased precede the NREM-REM transition less than 15 s[38]. Key regions contributing to the prediction accuracy of state transitions (Fig. 5e–h) in our LSTM RNN model may also be interesting for further research. Part of these regions have already been implicated in sleep modulation, such as ventrolateral

periaqueductal gray (vlPAG)[39], PGRN[39] and AMY[40] in promoting REM sleep, and GPe[41] and thalamus nucleus[42] in promoting NREM sleep. Other sensitive regions might also play vital roles in modulating the awake-sleep cycle, e.g., VTA, MnR. Previous studies have shown that a subset cells of VTA drive NREM and REM sleep through the lateral

hypothalamus[43] and fatal insomnia diseases are associated with raphe nuclei degeneration[44]. The remaining sensitive regions provide more candidates for further sleep research. Such long discriminate times and region-specific contribution demonstrated the global and sequential nature of sleep transitions, which further emphasized the need for a systematic whole-brain view to understand its fundamental mechanism.

Sleep-wake cycle is believed to be tightly regulated by a distributed network of sleep and wake promoting neurons[1,2], primarily located in subcortical regions. Recently, cortical regulation of sleep transition and maintenance has also been reported. Silencing neo-cortical layer 5 pyramidal neurons using SNAP25 knockout mouse decreased the cortico-subcortical communication and further markedly increased wakefulness[45]. In our study, we found widespread changes of cortical activities during "AW to NREM" and "NREM to AW" state transitions (Supplementary Fig. 7a, b), and some changes occurred earlier than those in subcortical regions, highlighting the important role of the cortex. Another notable example of cortical involvement in sleep regulation is retrosplenial cortex (RSP). Two recent studies in mice both showed that RSP was critically involved in REM sleep initiation and progression[46,47]. The current study also found RSP was highly activated during both NREM and REM state, and the activity of posterior DMN-like network (PC4 in Fig. 4a, e), largely overlapping with RSP, apparently preceded the "NREM to REM" state transition. Therefore, our study agrees well with recent progress on the cortical involvement in sleep-wake transitions.

Our MR-compatible LFP-fMRI setup enabled us to simultaneously record CA1 LFP signals and obtain good BOLD signals around the electrodes, including CA1 and RSP. As a characteristic event in hippocampus, SWRs have been shown to play an important role in memory consolidation[6] and coordinated hippocampal-thalamic-cortical communication[48]. Using NET fMRI approach, we provided the global view of SWR-evoked BOLD spatiotemporal pattern in NREM and AW states. Activation in RSP and hippocampus and deactivations in thalamus were consistent with previous results in macaque[22]. And optical imaging during natural sleep and anesthesia in mice also showed significant activation of RSP during SWRs[49]. Converging evidence, including the current results, clearly suggests the important role of RSP during SWRs, which may be related to the critical role of subiculum-retrosplenial pathway for SWR propagation from hippocampus to neocortex[50]. Interestingly, larger SWR-evoked BOLD responses in NREM state might attribute to the co-occurrence of spindle events (Fig. 7), as the spindle events showed whole-brain positive BOLD responses[51] (Fig. 8a and Supplementary Fig. 23, 24). In addition, we found the co-occurrence of SWRs and spindles elicited enhanced BOLD responses, compared to the sum of the responses of two solitary events (Fig. 8). Such effect was most notable in hippocampus, RSP and thalamus. Previous studies have shown that the interaction between thalamocortical spindles and hippocampal ripples promoted memory consolidation[4]. Interruption of the synchronization between ripple and spindle events appeared to interfere the efficiency of memory consolidation[52]. Thus, we speculate that the synergistic effects of SWRs and spindles might be related to the memory consolidation process, and further research is needed to examine the functional relevance of such phenomenon. Moreover, slow waves are thought to be critical for initiating information transfer between hippocampus and neocortex[5]. A previous human EEG-fMRI study[53] showed significant slow wave evoked activations in right parahippocampal gyrus, precuneus and posterior cingulate cortex, which were in good accordance with our results (Supplementary Fig. 25). It is known that the relationship among slow waves, spindles and SWRs is complex and may contribute to many cognition processes[54], especially memory consolidation[4,52]. Thus, the relationship among these events can be further explored, potentially using the current dataset. Our mouse sleep fMRI setup provides a tool for

investigating global spatiotemporal patterns of state dependent neural events and the mutual relationship of those events.

It is known that arousal may contribute to neuronal dynamics[55] and non-neuronal variations[56], e.g., vascular effects, head motion, and physiology, and they both contributed to fMRI dynamics. Various studies investigated the methods to remove arousal related non-neuronal variations, including global signal regression[56], data-driven approaches, e.g., ICA-FIX[57] and model based approaches, e.g., RETROICOR[58]. In our study, we applied a regression-based de-noising method modified from our previous studies[24,27], in which the "6 rp + 6 Δrp + 40 PCs" nuisance signals were used as regressors. The 40 PCs were derived from fMRI signals of non-brain tissues, largely capturing the non-neuronal signals[28], such as head motion, scanner drift, and physiological effects (Supplementary Fig. 2). Using the above regression approach, we believed that arousal induced non-neuronal nuisance effects on fMRI signals were largely suppressed.

Naturally, arousal fluctuations also modulate large scale brain activities[59], which has been shown to further contribute to resting-state fMRI dynamics[56]. Across different brain states in human, the network structure of spontaneous BOLD fluctuations was associated with that of slow electrophysiological activities[60]. Furthermore, arousal fluctuation synchronized the brain's functional systems through global wave propagations based on human fMRI and macaque ECoG[61]. Similar global wave propagation of infra-slow activity was shown in mouse based on the calcium and hemodynamic imaging in anesthetized and awake states[62]. Meanwhile, another human fMRI study showed different infra-slow propagation patterns between slow wave sleep and wakefulness[63]. Therefore, infra-slow brain activity across arousal states may orchestrate a wide range of interrelated neurophysiological and autonomic processes, and thus serve as a neural basis of low frequency spontaneous BOLD activity[64]. As arousal fluctuations are intricately linked to both neural and non-neural components in BOLD signals, the current mouse sleep fMRI setup may be advantageous to further investigate the neural basis of arousal related BOLD activity, with the simultaneous recording of electrophysiological and BOLD signals. Our current method of mouse sleep fMRI, and the open source data acquired using this method, features substantial arousal fluctuations from awake to REM sleep with simultaneously acquired cortical (ECoG) and hippocampus (LFP) signals. Therefore, our method and dataset provide an avenue to investigate the neural basis of arousal related fMRI dynamics.

In our study, there are several important limitations that need further improvement in the future. First, imaging coverage of deep brain regions was limited by the single loop receive coil used in our imaging setup. The signal-to-noise ratio can be further improved in deep brain regions, such as midbrain and pons, which are important for awake/sleep modulation[1]. Then, the sleep scoring criteria employed in the current study (Supplementary Fig. 3 and Supplementary Data 6) was largely adopted from previous mouse sleep studies[65] in the free-moving state. As no prior knowledge about ECoG/LFP and EMG signal characteristics during mouse sleep fMRI was known, the semi-automated sleep scoring approach developed here may be further refined. While we designed the 30° head holder to mimic the natural posture, sleep in the natural free-moving state may still be different from that in the head fixed state, which may result in different ECoG/LFP and EMG signal characteristics. Although other optical imaging studies of mouse sleep conducted in the head fixed condition also employed similar sleep scoring criteria[47,66], further detailed examination is much needed for improving sleep scoring in the head fixed state. Thus, optimization of receive coils is expected in the future for wider and better coverage of brain. In addition, the BOLD signal is the indirect measurement of neural activities based on neurovascular coupling[56]. Arousal fluctuation introduces physiological changes, e.g. respiration and heart rate, which could affect BOLD signal. In our fMRI preprocessing, the physiological nuisance could be

largely captured by PCs of the signals from the outside brain (Supplementary Fig. 2), in which the first 40 PCs showed significant correlation with respiration rate changes and were subsequently regressed out from fMRI data. Moreover, the significant correlation between electrophysiological and BOLD signals (Fig. 3d, e) indicated the potential neurophysiological relevance of NREM and REM state dependent BOLD activations. Nevertheless, as discussed above, such complex relationship between arousal related BOLD signal and neural activity is an important research direction itself, for better preprocessing strategies and more precise dissection of neural and non-neural contributions. Furthermore, recent advances in non-hemodynamic based fMRI such as direct imaging of neuronal activity (DIANA) methods[67], might eventually overcome such limitation.

In conclusion, we developed the mouse sleep fMRI method in un-anesthetized mice based on highly MR-compatible simultaneous electrophysiology at 9.4 T. For sleep research, this method provides an important avenue to investigate sleep dynamics at a global scale, and has great potential to integrate local and global view of sleep when further combined with other circuitry level tools in mice. For fMRI research, our method provides a convenient platform to examine the neural basis of the arousal related fMRI signal. Furthermore, our open source dataset of mouse sleep fMRI would provide a valuable source for both experimental and theoretical neuroscientists, and may help to establish a general multiscale framework of sleep from molecular, neuronal, circuitry and whole-brain levels.

## Methods

### Animals
Male wide type C57BL/6 J mice were obtained from Shanghai Laboratory Animal Center (Shanghai, China) at 8–10 weeks of age, weighted 20–30 g. Animals were group housed (5–6/cage) in the standard laboratory condition (temperature: 23 ±1 °C; humidity: 50–70%) under a 12 h light/dark cycle (light on from 7 a.m. to 7 p.m.) with food and water *ad libitum*. All animal experiments were approved by the Animal Care and Use Committee of the Institute of Neuroscience, Chinese Academy of Sciences, Shanghai, China.

### MR-compatible electrodes
Two types of MR-compatible electrodes were developed. First, an 18-contacts MR-compatible surface electrode was custom designed and manufactured for recording electrocorticogram (ECoG) and electromyography (EMG) signals (Fig. 1a). The array was made up of a flexible printed circuit (FPC) and two insulated gold wires, all soldered to an Omnetics 18-pin MR-compatible connector (A70242-001, Omnetics, USA, Minneapolis). The FPC was consisted of a 30 μm thin polyimide film, a 12 μm copper film, and a 3 μm gold-plated layer, with 14 ECoG recording site and two recording reference/ground (100 μm diameter; impedance, ~150 kOhm at 1KHz). Two insulated gold wires were used for EMG recording (50 μm diameter; impedance ~500 kOhm at 1 KHz).

Secondly, a MR-compatible depth electrode (Fig. 1b) was developed with 3 insulated gold wires, 2 bare silver wires and 1 graphene fiber (GF) electrode soldered to a flexible flat cable. One of three insulated gold wires (50 μm diameter; impedance, ~500 kOhm at 1 KHz) was used for recording ECoG, and the rest two for recording EMG. Two bare silver wires (100 μm diameter; impedance, ~60 kOhm at 1 kHz) were used as reference and ground. To record the hippocampal LFP signal, GF electrodes (impedance, ~60 kOhm at 1 kHz) were custom designed. GFs were fabricated through a one-step dimensionally confined hydrothermal process using suspensions of graphene oxide (GO) (monolayer, thickness: 0.8–1.2 nm; sheet diameter: 0.5–5 μm; #XF002-2, Nanjing/Jiangsu XFNANO Materials Technology, China). In a typical preparation, an 8 mg mL⁻¹ aqueous GO suspension was injected into a glass pipeline with a 0.9 mm inner diameter using a syringe. After being baked at 230 °C for 2 h with the two ends of the pipeline sealed, a GF matching the pipe geometry was produced. This preformed GF was then released from the pipeline by flow of N₂ and dried in air. The dried GF had a reduction in diameter to ~75 μm due to water loss and drying-induced alignment of the GO sheets. A GF with diameter of ~75 μm and length of 3 mm was connected to a bare copper wire with diameter of 100 μm using elargol. Parylene-C film of ~5 μm thickness was deposited onto the fibers in a custom made low-pressure coating system to finish the GF electrodes fabrication. Thus, the GF electrodes enabled unbiased fMRI mapping and excellent electrochemical performance, including low impedance and high electrical conductivity, which were not achievable by other electrodes.

### Surgical procedures
All surgical procedures were conducted under the standard aseptic condition. Mice were pretreated with 5 mg kg⁻¹ dexamethasone intraperitoneally 1 h before surgery to prevent brain edema. After anesthetized with isoflurane, mice were secured in a stereotaxic apparatus with a heating mat (mouseSTAT, Kent scientific cooperation). A midline sagittal incision was made along the scalp to expose the skull. The periosteum from the skull was removed by saline with cotton-tip applicator. After skull was dried out, a coat of self-etch adhesive (3 M ESPE Adper Easy One) was applied followed by light curing.

For MR-compatible ECoG electrode implantation, a sterile dental drill was used to drill off the surface of bone in the shape shown in Fig. 1a (length ~4 mm; width, ~6 mm; centered on bregma), with the bone around raphe retained. The window area was kept wet with saline, and the skull was slowly detached from the dura and raised at about 30° using a nasal stripper and tip tweezers. Gelatin sponge was applied to stop bleeding and keep the dura moist. Afterwards, the FPC part of the ECoG electrode was implanted epidurally with reference and ground sites attached on the bone of raphe. 2% sterilized agarose saline solution was filled between the skull and FPC. After the agarose was solidified, the window area was covered with light curing flowable dental resin to fix the FPC. Later, the two gold wires were implanted into posterior neck muscles for recording EMG. A head holder for awake imaging[26] was then attached on the skull above the cerebellum. Finally, dental cement was applied to smooth the surface of exposed skull. Mice were injected with meloxicam 0.5 mg kg⁻¹ (Baoding sunlight herb medicament, CN) subcutaneously for seven consecutive days post-surgery.

For MR-compatible depth electrode implantation, four 0.1 mm diameter holes were drilled in the skull. GF electrode was implanted into the dorsal CA1 region of the left hippocampus at stereotaxic coordinates AP = −1.94 mm, ML = −1.7 mm and DV = −1.15 mm. One gold wire was inserted epidurally at AP = +1 mm, ML = −1 mm to record ECoG, and the other two gold wire were implanted into posterior neck muscles to record EMG. Two bare silver wires were used as reference and ground, and inserted epidurally at ML = 0, AP = −5.5 mm and −6.5 mm, respectively. Light curing flowable dental resin was used for fixation of electrodes and head holder. Other procedures were identical as described above.

### Habituation
After seven-day recovery, mice were then habituated for fMRI for another seven days. Mice were head fixed on the animal bed with the recorded acoustic MRI scanning noise based on previous work[26]. Optical imaging studies usually record 3–5 h to obtain mice sleep more efficiently[29,46], as head fixed mice frequently fall asleep after 1.5 h and REM sleep frequently occurs after 2.5 h[46]. Therefore, 4 h head restraining was also utilized in our mouse sleep fMRI. The 30° head holder was designed to fit the natural sleep gesture of mice[29]. The animal bed was modified to allow more space for forelimb movement to reduce stress level[26]. The habituations were all carried out during 9a.m.–15p.m. with a fixed duration of 4 h and gradually increased noise levels. The detailed schedule was listed in Table 1. No reward was given during or after the habituation training.

**Table 1 | Summary of habituation schedules for awake mouse fMRI**

|  | day1 | day2 | day3 | day4 | day5 | day6 | day7 |
|---|---|---|---|---|---|---|---|
| Duration of Habituation | 240 min | 240 min | 240 min | 240 min | 240 min | 240 min | 240 min |
| Acoustic Noise | - | 60 dB | 60 dB | 90 dB | 90 dB | 110 dB | 110 dB |
| Ear plugs | - | - | - | - | - | + | + |

"–/+" denote absence or presence.

## Simultaneous neurophysiological recording and MRI acquisition

Neurophysiological signals were fed through a ZIF-CLIP anaglog headstage transmitter, PZ5 amplifier, RZ2 BioAmp processer, and finally recorded by the WS-8 workstation with Synapse software (all from Tucker-Davis Technologies, USA, Alachua). Neurophysiological signals were recorded at a sampling rate of 24414 Hz, high-pass filtered at 0.1 Hz and notch filtered at 50, 100 and 150 Hz, except for LFP signals, which were recorded without notch filter. PZ5 was connected to the external ground (waveguide tube of MRI) for stabilizing the signals. Respiration signal and MRI trigger signal were recorded at a sampling rate of 1024 Hz.

All MRI data were acquired with a Bruker BioSpec 9.4 T scanner (Software: ParaVision 6.0.1). An 86 mm volume coil was used for transmission and a single loop mouse head coil (Bruker, 1 cm diameter) was used for receiving. Mouse was head-fixed as described in previous section without using any anesthesia. A T2 weighted RARE anatomical image (TR: 3200 ms; TE: 34 ms; matrix size: 256 × 128; FOV: 18 × 9 mm²; slice thickness: 400 μm; resolution: 70 × 70 μm²) was acquired for coregistration purpose. After local shimming using Mapshim, functional images were acquired using single-shot echo planar imaging (EPI) with the following parameters: TR 2000 ms, TE 14 ms, FA 70°, matrix size 90 × 45, nominal in-plane resolution 200 × 200 μm², slice thickness 400 μm, slices number 22, 7200 EPI volumes (4 h). All 7200 volumes were acquired in a single EPI scan. Triggers were sent from the MRI console for each slice acquisition and recorded along with the electrophysiological signal.

## Electrophysiological signal processing

The off-line correction of MRI gradient artifacts in the electrophysiological signals was conducted using the fMRI Artifact Slice Template Removal algorithm (FASTR)[68] (Supplementary Fig. 1). Briefly, the imaging artifact waveforms were segmented, averaged and iteratively subtracted from the raw electrophysiological signals, according to the concurrently acquired trigger signal from the MRI scanner. This procedure was performed through FMRIB in EEGLAB (https://fsl.fmrib.ox.ac.uk/eeglab/fmribplugin/). After careful visual inspection, we found a few noisy ECoG channels in a small number of scans. Thus, we interpolated the noisy ECoG signal by weighted averaging signals from other good ones, similar to the previous study[69] using the neighbor interpolation method (Supplementary Fig. 1 and Supplementary Data 5). Considering there were $k$ ($k < 4$) bad channels in 14 channels, the weighted averaging method could be simply formulated as

$$S^{k_i} = \frac{1}{14-k}\sum_{j=0}^{14-k}\left(Dis_{k_i-good}\right)^{\lambda}\cdot S^{j}_{good} \qquad (1)$$

where $S^{k_i}$, $S_{good}$ and $Dis_{k_i-good}$ were the ECoG signals of the $k_i$th interpolated channel, all good channels and the corresponding Euclidean distance, and $\lambda$ represented the exponential constraints ($\lambda < 0$) using the weighted averaging method. Parameter $\lambda$ was estimated using other scans without noisy ECoG channels based on the least square fitting. Negative $\lambda$ indicated farther good channels contributed lower weights on the interpolation of $k_i$th noisy channel. The denoised

electrophysiological signals were further down-sampled to 1024 Hz for subsequent analysis.

For brain state classification[65], one channel within each session was selected for further analysis, and the selected channels were listed in the Supplementary Data 6. Then, we calculated the power spectrum for the ECoG/iEEG and EMG (160–250 Hz) data with 3 s sliding windows and 1 s step size, using the multi-taper method implemented in Chronux (http://chronux.org/). Next, we computed the theta (6–12 Hz) and delta (1–4 Hz) power and theta/delta power ratio, which were further smoothed using "medfilt1" (20 points) in MATLAB. For each session, we used the following criteria for tentatively defining brain states: (1) a time point was classified as NREM sleep if the smoothed delta power was higher than its mean; (2) a time point was assigned as REM sleep if the smoothed theta/delta power ratio was two standard deviations higher than its mean and the EMG power was one standard deviation lower than its mean; and (3) all remaining time points were classified as AW state. Then, we further manually adjusted the classification of brain states as following: (1) For NREM state, according to the ECoG power spectrum, we adjusted the start or end point to the point with the greatest ascent or descent speed of smoothed delta power. For REM state, the start point was adjusted to the end point of the previous NREM, and the end point was adjusted to the point with the greatest descent speed of smoothed theta power. If the greatest ascent or descent point of EMG power was different from that of ECoG/iEEG signals, the midpoint between the two was defined as the transition point; (2) For sessions with noisy EMG recordings, we used the head motion (framewise displacement) estimated from fMRI data as substitute for EMG power; and (3) epochs with short durations (<5 s) were manually merged to the nearest sleep stage. An example of the above classification procedure (session 1: channel 10, 11500–13000 s) was shown in Supplementary Fig. 3.

Spindle and SWRs events were identified in the LFP dataset based on previous studies[5,7]. To identify the spindle events, raw iEEG signals in mPFC were bandpass filtered (10–16 Hz) with Butterworth filter. A spindle event was identified if the envelope of the filtered iEEG signal was larger than its mean + 1.5 s.d. in NREM state (Supplementary Fig. 16). To identify the SWRs events, raw LFP signals in CA1 of hippocampus were bandpass filtered (120–250 Hz) with Butterworth filter. A SWRs event was detected if (1) the envelope of the filtered LFP signal larger than its mean + 3 s.d. and (2) the power of the filtered LFP signal larger than its temporal mean. The center of spindle or SWRs event was defined as the time of maximum peak of the threshold-passed envelope. The beginning and the end (i.e., event duration) was measured before and after this maximum peak when the amplitude dropped below the mean value of the corresponding envelope (Supplementary Fig. 16). Only spindles with 0.4–3 s durations were included. Slow waves were identified based on the procedures described previously[5]. The raw signal was first down-sampled to 1024 kHz. Then, for slow waves detection, mPFC iEEG signals were filtered between 0.3 and 4.5 Hz with a two-order Butterworth bandpass filter. A slow wave was detected in NREM state if the following three criteria were all fulfilled: (1) the interval (T) of negative wave between 0.4 and 2.0 s; (2) top 35% negative amplitude (N) and (3) top 45% negative-to-positive peak-to-peak amplitude (M). Slow wave onset and offset was defined by the time of the first and third zero crossing, respectively (Supplementary Fig. 25).

## fMRI data preprocessing

All subsequent procedures were performed using custom scripts in MATLAB 2020a (MathWorks, Natick, MA) and SPM12 (http://www.fil.ion.ucl.ac.uk/spm/). The mouse brain was extracted manually using ITK-SNAP (http://www.itksnap.org/). First, each fMRI scan was slice-timing corrected and registered to the scan-specific structural image using rigid body transformation and the scan-specific structure image was then nonlinearly transformed to a study-specific mouse template (https://atlas.brain-map.org/) for group analysis. Then, a light spatial

smoothing (0.4 mm isotropic Gaussian kernel) was performed but no band-pass filter was applied to the BOLD time series. Furthermore, to minimize the effects of scanner drift, motion and other non-neural physiological noises, BOLD signals were regressed by "6 rp + 6 Δrp + 40 PCs" nuisance signals[24,25] (Supplementary Fig. 2a–h). "6 rp + 6 Δrp" nuisance signals represented 6 head motion parameters and their 1st order first derivatives, and "40 PCs" were the first 40 principal components from the BOLD signals of non-brain tissue, e.g., the muscles. The regression-based denoising strategy was adopted from a previous study[25], in which the PCs estimated from tissues outside the brain were used to model non-neural signal variations and produced a moderate improvement in specificity. The Pearson's correlation coefficients between the frame-wise displacement (FD) and DVARS (D referring to temporal derivative of time courses, VARS referring to RMS variance over voxels) were calculated to quantitatively reflect the extent to which the motion related signal was reduced by given regressors (Fig. 2d).

## General linear model of brain state activation and electro-physiological validation

GLM based statistical analysis was conducted using the mouse-specific HRF from our previous study[25], in which NREM and REM states were set as the predictors and thus the AW state was used implicitly as the baseline. Standard first level analysis was done for individual EPI scans. For second level analysis, flexible factorial model (brain states and mouse individuals) was conducted to generate the activation maps with FDR corrected $p < 0.05$. Furthermore, to investigate the electro-physiological relevance of BOLD signal variations in NREM and REM states, we estimated the Pearson's correlation coefficients between relative ECoG (or LFP) powers (vs. AW state) and corresponding relative BOLD amplitudes.

## Group principal component analysis (PCA) of BOLD fMRI signal

Each individual EPI scan comprising $t$ time points and $v$ voxels could be represented as a 2-dimensional time-space matrix ($S_{t \times v}$). In the initial scan-level PCA step, fMRI data of each scan was reduced to $p$ components of dimension ($W_{p \times v}$, $p \ll t$). In the second PCA step, group PCA was performed on the concatenated data ($W_{pN \times v}$) from all scans, in which data from $N$ scans were stacked along the reduce dimension. Then, the group spatial PCs ($W_{p_0 \times v}$) were back-projected to raw data ($S_{t \times v}$) to reconstruct the time courses of each component (tPCs: $S_{t \times p_0}$) for each individual EPI scan.

## Brain state prediction using LSTM RNNs

The long short-term memory (LSTM) recurrent neural networks (RNNs) model was built to predict the brain state based on its functional profile, i.e., tPCs, and their temporal dependency of BOLD signals on its preceding time points. Given the $L$ most recent timesteps $\{X_{t-L+1}, X_{t-L+2}, \ldots, X_t\}$, the goal at timestep $t$ was to predict the state of M timesteps into the future, $\hat{X}_{t+M}$. The architecture of the LSTM RNNs used in this study was illustrated in Fig. 5a, including one (or two or three) hidden LSTM layer(s) and one fully connected layer. Multiple hidden LSTM layers could be used to encode the functional information with temporal dependency for each time point, and the fully connected layer was used to learn a mapping between the learned feature representation and brain states. The functional representation encoded in each LSTM layer was calculated as

$$f_t^l = \sigma\left(W_f^l \cdot \left[h_{t-1}^l, x_t^l\right] + b_f^l\right) \tag{2}$$

$$i_t^l = \sigma\left(W_i^l \cdot \left[h_{t-1}^l, x_t^l\right] + b_i^l\right) \tag{3}$$

$$\widetilde{C}_t^l = \tanh\left(W_C^l \cdot \left[h_{t-1}^l, x_t^l\right] + b_C^l\right) \tag{4}$$

$$C_t^l = f_t^l {}^* C_{t-1}^l + i_t^l {}^* \widetilde{C}_t^l \tag{5}$$

$$o_t^l = \sigma\left(W_o^l \cdot \left[h_{t-1}^l, x_t^l\right] + b_o^l\right) \tag{6}$$

$$h_t^l = o_t^l {}^* \tanh\left(C_t^l\right) \tag{7}$$

where $f_t^l, i_t^l, C_t^l, h_t^l$ and $x_t^l$ denoted the output of forget gate, input gate, cell state, hidden state and the input feature vector of the $l$-th LSTM layer ($l = 1$, 2 or 3) at the $t$-th time point, respectively. In addition, $\sigma$ represented the sigmoid function and $W^l$ and $b^l$ denoted the gates weights and biases of the $l$-th LSTM layer. A fully connected layer with $s$ output nodes was utilized for predicting the brain state as

$$s_t = softmax\left(W_s \cdot h_t^2 + b_s\right) \tag{8}$$

where $s$ was the number of brain state to be predicted, and $h_t^2$ was the hidden state output of the last LSTM layer which encoded the input functional signature at the $t$-th time point and the temporal dependency information encoded in the cell state from its preceding time points. Softmax cross-entropy between empirical and predicted brain states was used as the objective function to optimize the LSTM RNNs model.

Based on the abovementioned group PCA results, the time series of each principal component (tPCs) were normalized to $z$ scores, and then used as the input of the LSTM RNNs model to predict their corresponding brain states. We chose the first 100 tPCs for characterizing the functional profiles of BOLD signal dynamics along with brain state changes. Firstly, we divided the mouse brain states into 7 prediction categories, including (1) AW only, (2) NREM only, (3) REM only, (4) AW to NREM, (5) NREM to AW, (6) NREM to REM and (7) REM to AW state. Then, we utilized the bootstrap method to avoid the prediction bias from highly unbalanced samples of brain states, e.g., a low percentage of REM state and REM related state transitions. Thus, we randomly selected the BOLD clips (100 tPCs × 120 s) with same sample sizes for 7 categories, which was the minimum number of BOLD clips among 7 prediction categories. The resulting clips were used as the input features of our LSTM RNNs model. Finally, to minimize the prediction bias underlying the sampling procedure, we conducted the bootstrap method for 500 times and repeated the predictions using the correspondingly 500 groups of BOLD clips. For each group of BOLD clips, a 10-fold cross-validation was carried out to improve the robustness of the prediction performance.

Particularly, we adopted the "adaptive moment estimation (ADAM)" optimizer with a learning rate of 0.01, which was updated every 1000 training steps with a decay rate of 0.5, and the total number or training steps was set to 8000. Batch size was set to 128 during the training procedure. Parameters including number of hidden layers (1, 2 and 3) and number of nodes in hidden layers (25, 50, …, 200) were selected based on their prediction performance on the validation dataset. The parameter selection was performed on empirical dataset, and the selected parameters were used for the null dataset without further optimization. The null dataset was constructed by randomly shuffling the predicted categories for each group of BOLD clips. Thus, the discriminate time of brain state transition was defined as the farthest time point when prediction accuracy from the empirical dataset was out of the 95% confidence interval (CI) of that from the null dataset.

To further reveal which brain region(s) contributed to the prediction most, we carried out a sensitivity analysis based on our 500 times bootstrap sampled dataset. The sensitivity analysis was conducted by evaluating how changes of the 100 tPCs affected the prediction accuracy. Briefly, with the trained LSTM RNNs model remaining unchanged, time courses of 100 PCs were set to 0 one by one from the model input. Changes ($\triangle ACC_{p_0}$) in the prediction accuracy were then

reconstructed by multiplying the spatial weights matrix ($W_{p_0 \times v}$) of group PCA. The resulting spatial weighted accuracy ($\triangle ACC_{p_0} \cdot W_{p_0 \times v}$) was defined as the sensitivities of LSTM RNNs model on brain state prediction.

## Reporting summary

Further information on research design is available in the Nature Portfolio Reporting Summary linked to this article.

## Data availability

The manuscript released a simultaneous electrophysiological-fMRI dataset, which is available at https://doi.org/10.12412/BSDC.1668502646.20001. The source data underlying Figs. 2–8 and Supplementary Figs. 3–5, 13 and 20 are provided as a Source Data file. Source data are provided with this paper.

## Code availability

Codes used in this study are available at https://github.com/TrangeTung/Mouse_Sleep_fMRI.

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

## Acknowledgements

The authors thank Zhe Zhang (ION, CAS) and Min Xu (ION, CAS) for helpful discussions, Shou Qiu, Feikai Lin and Chun Xu (ION, CAS) for help in hippocampal CA1 recording, and Brain Science Data Center, Chinese Academy of Sciences for dataset release. This work was supported by the National Science and Technology Innovation 2030 Major Program (2021ZD0202204 to XD, 2021ZD0200100 to ZL), Strategic Priority Research Program of Chinese Academy of Sciences (XDBS01030100 to ZL), Shanghai Municipal Science and Technology Major Project (2018SHZDZX05 to ZL), the National Natural Science Foundation of China (82171899 to ZL, U21A6005 to YF), Lingang Laboratory (LG202104-02-06 to ZL), Key-Area Research, Development Program of Guangdong Province (2018B030340001 and 2018B030333001 to YF) and Science and Technology Commission of Shanghai Municipality (201409002100 to FF). Cartoon mouse images in the manuscript were drawn in Figdraw (https://www.figdraw.com/).

## Author contributions

C.T., X.D. and Z.L. designed and supervised the study; Y.Q., G.L., X.D. and Z.L. designed the MR-compatible electrodes; Y.Q., Y.Y., K.Z., B.B., M.P., J.Y., G.T., and C.T. collected the electrophysiological and fMRI data; Y.Y., Y.Q., C.T. and Z.L. conducted the data analysis. C.T., Y.Y., Y.Q., X.D. and Z.L. wrote the original draft and revised the draft. All authors discussed and commented on the paper.

## Competing interests

The authors declare no competing interests.

## Additional information

**Peer review information** : *Nature Communications* thanks Marcus Raichle and the other, anonymous, reviewer for their contribution to the peer review of this work. Peer reviewer reports are available.

