## [Peer Review File · Nature Communications]

Sleep fMRI with simultaneous electrophysiology at 9.4T in male miceREVIEWER COMMENTS

Reviewer #1 (Remarks to the Author):

Nature Communications manuscript NCOMM-22-50604-T

Title: Mouse sleep fMRI with simultaneous electrophysiology

This is a fascinating manuscript presenting a treasure trove of interesting new brain data on sleep. The data come from mice who were, remarkably, able to sleep in a MRI scanner with an array of specially designed, non-MRI interfering electrodes implanted in their brain. The utility of the model and the applicability of the data to our understanding sleep in other mammalian species including humans is very promising. In addition, the authors are quite explicit in their willingness to share these data with other researchers. Given the technical sophistication of the experimental model used, this is an especially important component of this research from the perspective of the neuroscience community at large.

Another important feature of this work is the way they frame their approach. By combining implanted brain electrodes with MRI, the authors simultaneously address both the global and the local features of the mammalian brain as it changes from WAKE to NREM and REM sleep and then back to WAKE. These include not only the unique features of these 3 brain states but also the features of the transitions from one state to another. It is a forward-looking approach as we come to realize the importance of integrating our understanding of both the micro and macro features of the brain function.

My main concern with this manuscript, in its present form, is the paucity of reference to and apparent understanding of existing information in the literature especially regarding the fMRI BOLD signal and its relationship to the underlying neurophysiology and brain metabolism. It is insufficient to state (line 396) "It is well known that BOLD fMRI is a complex combination of neural, physiological, and vascular combinations, and such complexity is particularly acute in "resting-state" fMRI" without a single reference. For example, work in humans [1] as well as mice [2] has clearly shown to be related to what has come to be called infra-slow activity (ISA). ISA has a remarkably long but often overlooked history in neurophysiology (e.g., see [3] for an excellent review). Our understanding of ISA has continued to expand as the role of traveling waves has entered the picture (e.g., see [4]) and state changes (e.g., WAKE to NREM sleep) which are associated with directional changes in these traveling waves [5]. How might their work relate to this story?

The authors express concerns about the role of arousal and its role in the interpretation of their results (see lines 401 and beyond) without clearly expressing how that might occur and how it might be monitored (e.g., pupillary diameter or heart rate variability) or dealt with. There are no references to current work on the subject. I would encourage them to have a look at the work of McCormick and his colleagues (e.g., see Figure 2 in [6], which relates to the work in this manuscript).

Finally, the authors bring up the issue of state transitions without mention of work done in this area in the awake state (e.g., see [7, 8]). The loss of state transition activity in patients with Alzheimer's disease is strikingly apparent (see Figure 1D in [9]). It would seem to me that the authors are in a unique position to relate their sleep findings into a broader context of how the brain switches states whether awake or asleep. Future work might include experiments with transgenic mice who are programmed to develop Alzheimer's disease.

In summary, I enjoyed reading this paper and commend the authors on a technically sophisticated approach to a further our understanding of the sleeping mammalian brain and their willingness to share these interesting data. As I see it, the challenge remaining for them is more adequately relating their work to what we already know.

Marcus E. Raichle, M.D

1. He, B.J., et al., Electrophysiological correlates of the brain's intrinsic large-scale functional architecture. *Proc Natl Acad Sci U S A*, 2008. 105(41): p. 16039-44.
2. Mitra, A., et al., Spontaneous Infra-slow Brain Activity Has Unique Spatiotemporal Dynamics and Laminar Structure. *Neuron*, 2018.
3. Palva, J.M. and S. Palva, Infra-slow fluctuations in electrophysiological recordings, blood-oxygenation-level-dependent signals, and psychophysical time series. *Neuroimage*, 2012.
4. Raut, R.V., et al., Global waves synchronize the brain's functional systems with fluctuating arousal. *Sci Adv*, 2021. 7(30).
5. Mitra, A., et al., Propagated infra-slow intrinsic brain activity reorganizes across wake and slow wave sleep. *Elife*, 2015. 4.
6. McGinley, M.J., S.V. David, and D.A. McCormick, Cortical Membrane Potential Signature of Optimal States for Sensory Signal Detection. *Neuron*, 2015. 87(1): p. 179-92.
7. Fox, M.D., et al., Transient BOLD responses at block transitions. *Neuroimage*, 2005. 28(4): p. 956-66.
8. Hayden, B.Y., D.V. Smith, and M.L. Platt, Electrophysiological correlates of default-mode processing in macaque posterior cingulate cortex. *Proc Natl Acad Sci U S A*, 2009. 106(14): p. 5948-53.
9. Lustig, C., et al., Functional deactivations: change with age and dementia of the Alzheimer type. *Proc Natl Acad Sci U S A*, 2003. 100(24): p. 14504-9.

Reviewer #2 (Remarks to the Author):

The study by Yu et al uses 9.4T fMRI in combination with EEG and hippocampal LFP recordings to quantify sleep associated changes of the fMRI BOLD signal. The authors developed electrodes that allowed them acquiring electrophysiological data while simultaneously imaging the BOLD signal in mice. They quantified sleep stage specific changes in BOLD signals for diverse brain areas as well as BOLD changes during the transitions between wakefulness, NREM and REM sleep. Furthermore, they used principal component analysis (PCA) on BOLD signals to extract components that discriminate best between WAKE, NREM and REM sleep. Then they used the results from PCA as input to neural networks that allowed them to classify sleep stage transitions based on the BOLD principal components. Finally, they investigated BOLD responses relative to hippocampal sharp wave ripples that either occurred alone or coupled to spindles.

The work involves a diversity of cutting-edge methods to quantify sleep stage specific changes in BOLD responses and I very much appreciate that the authors provide easy access to the raw and preprocessed data. However, the current version of the manuscript lacks important information about sleep scoring procedures that are particularly relevant for this study. Additionally after checking some of the available

data one could clearly identify episodes that were classified as REM sleep but do not fulfill established criteria for such classification. Additionally, the data often shows very strong artifacts and it is not clear whether these were excluded from analysis or not. Besides this technical limitation, I see a major weakness deriving from the poor conceptual embedding and elaboration on the functional findings in the Introduction and Discussion. The authors mainly focus on technical rather than physiological and conceptual aspects. Nevertheless the authors might consider the following points:

Major points:

It is not clear how the authors performed sleep scoring. How did they score 1 s epochs? The reference they refer for established criteria is not describing any details for sleep scoring and therefore does not provide the necessary information to judge about the authors approach. Please, provide in depth information on these procedures. Especially, if sleep classification was (semi-) automated or manual. The available data clearly shows that sleep staging does not follow established criteria and must be checked very carefully.

The authors often imply causality when referring to correlational analysis. E.g., line 197-200: "For REM state, significant correlation between theta band-limited power and BOLD changes was observed (Fig. 3d, e, right panels), suggesting the contribution of theta band on BOLD changes."

The authors refer to spindle band for frequency bands ranging from 7 - 37 Hz (line 260). Sleep spindles do not reach such high frequencies! This also relates to figure 6 a. The authors generally should indicate the time window for which they find significant differences for these frequencies, and the exact method they used for statistical testing.

The analysis shown in figure 7 lacks the BOLD signals for solitary spindles as a control. Previous work showed that spindles are generally accompanied by a widespread increase, e.g., in cortical activity. Currently, it is not clear whether it is actually the co-occurrence of ripples and spindles that drives this wide spread increase in the BOLD signal or if spindles generally show such activity pattern. On a related note, the interpretation that coupled spindle events "contribute" to increased BOLD responses evoked by SWRs is at best unclear: BOLD activity elicited by spindles not coupled to SWRs would have to be subtracted to substantiate this claim, and the combined events are likely more than the sum of their parts, both in terms of the neural activities elicited and the functions subserved.

One important omission that would allow for better embedding of the present findings into existing models of sleep is that the authors did not investigate/explore BOLD activity associated with slow oscillations which are thought to play a central role in initiating information transfer between hippocampus and neocortex.

In general the readability of the manuscript can be improved.

Minor issues:

- l.160: The information contained in Suppl. Fig. 2 should be briefly spelled out here, to allow readers to grasp the basic procedures of MR preprocessing.

- l.182: The argument that transitions between sleep stages occur more frequently is valid, but it over-emphasizes the notion of sleep stages as stationary, potentially at the cost of dynamic microstructure (which the authors do investigate with NET-fMRI).

- l.226: I found the term "non-reversible transitions" used here and elsewhere unclear – perhaps 'non-commutative' or 'asymmetric' would be better.

- l.517: As far as I see, the habituation procedure has not been explicitly described before.

- I.530: Why did the authors apply the wide notch filter between 100 and 150 Hz?
- I.546: In humans, the most problematic artifact with simultaneous EEG-fMRI is not due to imaging, but to pulse-induced electrode movements in the static magnetic field. I imagine these movements are considerably smaller in the present case. Nevertheless, the authors should mention if and how this potential source of artifacts was taken into account.
- I.557: The authors should clarify the range and mean number of electrodes interpolated across animals. It should also be clarified how many of these were located at the border of the ECoG arrays, as interpolation results are likely less reliable in these cases.
- I.609: Is there any literature on potential differences between HRFs in the mouse across different brain states?
- I.659: How did the authors determine the cut-off of 100 temporal principal components?
- Out of curiosity: can the authors say anything about spindle-associated cerebellar activity? (cf. Xu, W., De Carvalho, F., Clarke, A. K., & Jackson, A., 2021, Progress in Neurobiology)

REVIEWER COMMENTS

Reviewer #1 (Remarks to the Author):

Nature Communications manuscript NCOMM-22-50604-T

Title: Mouse sleep fMRI with simultaneous electrophysiology

This is a fascinating manuscript presenting a treasure trove of interesting new brain data on sleep. The data come from mice who were, remarkably, able to sleep in a MRI scanner with an array of specially designed, non-MRI interfering electrodes implanted in their brain. The utility of the model and the applicability of the data to our understanding sleep in other mammalian species including humans is very promising. In addition, the authors are quite explicit in their willingness to share these data with other researchers. Given the technical sophistication of the experimental model used, this is an especially important component of this research from the perspective of the neuroscience community at large.

Another important feature of this work is the way they frame their approach. By combining implanted brain electrodes with MRI, the authors simultaneously address both the global and the local features of the mammalian brain as it changes from WAKE to NREM and REM sleep and then back to WAKE. These include not only the unique features of these 3 brain states but also the features of the transitions from one state to another. It is a forward-looking approach as we come to realize the importance of integrating our understanding of both the micro and macro features of the brain function.

1. My main concern with this manuscript, in its present form, is the paucity of reference to and apparent understanding of existing information in the literature especially regarding the fMRI BOLD signal and its relationship to the underlying neurophysiology and brain metabolism. It is insufficient to state (line 396) “It is well known that BOLD fMRI is a complex combination of neural, physiological, and vascular combinations, and such complexity is particularly acute in “resting-state” fMRI” without a single reference. For example, work in humans [1] as well as mice [2] has clearly shown to be related to what has come to be called infra-slow activity (ISA). ISA has a remarkably long but often overlooked history in neurophysiology (e.g., see [3] for an excellent review). Our understanding of ISA has continued to expand as the role of traveling waves has entered the picture (e.g., see [4]) and state changes (e.g., WAKE to NREM sleep) which are associated with directional changes in these traveling waves [5]. How might their work relate to this story?

Response:

Thank you for your insightful advice. We significantly expanded the discussion regarding the current research on neural basis of resting state fMRI at different brain states, including all references in your comment (page 17, line 32) as follows:

“Naturally, arousal fluctuations also modulate large scale brain activities¹, which has been shown to further contribute to resting-state fMRI dynamics². Across different brain states in human, the network structure of spontaneous BOLD fluctuations was associated with that of slow electrophysiological activities³. Furthermore, arousal fluctuation synchronized the brain’s functional systems through global wave propagations based on human fMRI and macaque ECo

G⁴. Similar global wave propagation of infra-slow activity was shown in mouse based on the calcium and hemodynamic imaging in anesthetized and awake states⁵. Meanwhile, another human fMRI study showed different infra-slow propagation patterns between slow wave sleep and wakefulness⁶. Therefore, infra-slow brain activity across arousal states may orchestrate a wide range of interrelated neurophysiological and autonomic processes, and thus serve as a neural basis of low frequency spontaneous BOLD activity⁷. As arousal fluctuations are intricately linked to both neural and non-neural components in BOLD signal, the current mouse sleep fMRI setup may be advantageous to further investigate the neural basis of arousal related BOLD activity, with the simultaneous recording of electrophysiological and BOLD signal.”

2. The authors express concerns about the role of arousal and its role in the interpretation of their results (see lines 401 and beyond) without clearly expressing how that might occur and how it might be monitored (e.g., pupillary diameter or heart rate variability) or dealt with. There are no references to current work on the subject. I would encourage them to have a look at the work of McCormick and his colleagues (e.g., see Figure 2 in [6], which relates to the work in this manuscript).

Response:

We really appreciated the reviewer’s constructive comment. In the current study, we mainly investigated the dynamics of awake-sleep cycle, which was classified using the gold standard electrophysiological signals. We did not monitor the pupillary diameter or heart rate to infer the arousal fluctuations¹ as the complexity of simultaneous Ephys.-fMRI setup was already highly demanding for both human experimenters and animals. However, in our dataset, the arousal fluctuation could be directly inferred from the Ephys. signals by taking the ratio of power in beta- and theta- range frequency bands (12-25 Hz and 3-7 Hz, respectively)^{8,9}.

Now we revised our manuscript to better discuss the role of arousal and its role in the interpretation of our results (Page 17, line 20), as follows:

“It is known that arousal may contribute to neuronal dynamics¹⁰ and non-neuronal variations², e.g., vascular effects, head motion, and physiology, and they both contributed to fMRI dynamics. Various studies investigated the methods to remove arousal related non-neuronal variations, including global signal regression², data-driven approaches, e.g., ICA-FIX¹¹ and model based approaches, e.g., RETROICOR¹². In our study, we applied a regression-based de-noising method modified from our previous studies^{13,14}, in which the “6 rp + 6 Δrp + 40 PCs” nuisance signals were used as regressors. The 40 PCs were derived from fMRI signals of non-brain tissues, largely capturing the non-neuronal signals¹⁵, such as head motion, scanner drift, and physiological effects (Now Supplementary Fig. 2). Using the above regression approach, we believed that arousal induced non-neuronal nuisance effects on fMRI signals were largely suppressed.

Naturally, arousal fluctuations also modulate large scale brain activities¹, which has been shown to further contribute to resting-state fMRI dynamics². Across different brain states in human, the network structure of spontaneous BOLD fluctuations was associated with that of slow electrophysiological activities³. Furthermore, arousal fluctuation synchronized the brain’s functional systems through global wave propagations based on human fMRI and macaque ECoG⁴. Similar global wave propagation of infra-slow activity was shown in mouse based on the calcium and hemodynamic imaging in anesthetized and awake states⁵. Meanwhile, another

human fMRI study showed different infra-slow propagation patterns between slow wave sleep and wakefulness⁶. Therefore, infra-slow brain activity across arousal states may orchestrate a wide range of interrelated neurophysiological and autonomic processes, and thus serve as a neural basis of low frequency spontaneous BOLD activity⁷. As arousal fluctuations are intricately linked to both neural and non-neural components in BOLD signal, the current mouse sleep fMRI setup may be advantageous to further investigate the neural basis of arousal related BOLD activity, with the simultaneous recording of electrophysiological and BOLD signal.”

3. Finally, the authors bring up the issue of state transitions without mention of work done in this area in the awake state (e.g., see [7, 8]). The loss of state transition activity in patients with Alzheimer’s disease is strikingly apparent (see Figure 1D in [9]). It would seem to me that the authors are in a unique position to relate their sleep findings into a broader context of how the brain switches states whether awake or asleep. Future work might include experiments with transgenic mice who are programmed to develop Alzheimer’s disease.

Response:

Thank you for your nice advice. Our original manuscript only discussed wake/sleep state related transitions, and now we added more discussions about the state dynamics in a broader definition (Page 15, line 6), as follows:

“The dynamics of broadly defined state transition and fluctuations in resting-state fMRI is now widely investigated, and has been implicated in cognitive processes^{16, 17} and brain disorders, e.g. Alzheimer’s disease (AD)¹⁸ and obsessive–compulsive disorder (OCD)¹⁹. The functional network of those patients with brain disorders exhibited abnormal dynamic rhythms^{18, 19}, indicating potential clinical relevance. And also, consciousness has been shown to modulate the diversity of the state dynamics across different sedation levels²⁰. Utilizing the rich transgenic mouse disease models, e.g., various AD mouse models, future research can be conducted based on our mouse sleep fMRI setup to further investigate the mechanisms of state transition dynamics and its role in brain disorders.”

In summary, I enjoyed reading this paper and commend the authors on a technically sophisticated approach to a further our understanding of the sleeping mammalian brain and their willingness to share these interesting data. As I see it, the challenge remaining for them is more adequately relating their work to what we already know.

Marcus E. Raichle, M.D

1. He, B.J., et al., Electrophysiological correlates of the brain's intrinsic large-scale functional architecture. *Proc Natl Acad Sci U S A*, 2008. 105(41): p. 16039-44.
2. Mitra, A., et al., Spontaneous Infra-slow Brain Activity Has Unique Spatiotemporal Dynamics and Laminar Structure. *Neuron*, 2018.
3. Palva, J.M. and S. Palva, Infra-slow fluctuations in electrophysiological recordings, blood-oxygenation-level-dependent signals, and psychophysical time series. *Neuroimage*, 2012.
4. Raut, R.V., et al., Global waves synchronize the brain's functional systems with fluctuating arousal. *Sci Adv*, 2021. 7(30).
5. Mitra, A., et al., Propagated infra-slow intrinsic brain activity reorganizes across wake and slow wave sleep. *Elife*, 2015. 4.
6. McGinley, M.J., S.V. David, and D.A. McCormick, Cortical Membrane Potential Signature

- of Optimal States for Sensory Signal Detection. *Neuron*, 2015. 87(1): p. 179-92.
7. Fox, M.D., et al., Transient BOLD responses at block transitions. *Neuroimage*, 2005. 28(4): p. 956-66.
8. Hayden, B.Y., D.V. Smith, and M.L. Platt, Electrophysiological correlates of default-mode processing in macaque posterior cingulate cortex. *Proc Natl Acad Sci U S A*, 2009. 106(14): p. 5948-53.
9. Lustig, C., et al., Functional deactivations: change with age and dementia of the Alzheimer type. *Proc Natl Acad Sci U S A*, 2003. 100(24): p. 14504-9.

Reviewer #2 (Remarks to the Author):

The study by Yu et al uses 9.4T fMRI in combination with EEG and hippocampal LFP recordings to quantify sleep associated changes of the fMRI BOLD signal. The authors developed electrodes that allowed them acquiring electrophysiological data while simultaneously imaging the BOLD signal in mice. They quantified sleep stage specific changes in BOLD signals for diverse brain areas as well as BOLD changes during the transitions between wakefulness, NREM and REM sleep. Furthermore, they used principal component analysis (PCA) on BOLD signals to extract components that discriminate best between WAKE, NREM and REM sleep. Then they used the results from PCA as input to neural networks that allowed them to classify sleep stage transitions based on the BOLD principal components. Finally, they investigated BOLD responses relative to hippocampal sharp wave ripples that either occurred alone or coupled to spindles.

The work involves a diversity of cutting-edge methods to quantify sleep stage specific changes in BOLD responses and I very much appreciate that the authors provide easy access to the raw and preprocessed data. However, the current version of the manuscript lacks important information about sleep scoring procedures that are particularly relevant for this study. Additionally after checking some of the available data one could clearly identify episodes that were classified as REM sleep but do not fulfill established criteria for such classification. Additionally, the data often shows very strong artifacts and it is not clear whether these were excluded from analysis or not. Besides this technical limitation, I see a major weakness deriving from the poor conceptual embedding and elaboration on the functional findings in the Introduction and Discussion. The authors mainly focus on technical rather than physiological and conceptual aspects. Nevertheless the authors might consider the following points:

Response:

We really appreciate the reviewer's constructive comments. We provided detailed responses for technical points below.

Regarding the "major weakness deriving from the poor conceptual embedding and elaboration on the functional findings", we sincerely apologize for such weakness. Indeed as the reviewer already pointed out, this manuscript mainly focuses on the method development, which is highly sophisticated and technically challenging. Therefore, to facilitate further data mining by other researchers, we provided the open-accessible dataset with detailed brain state and neural event labels. We believe much more information can be extracted from this dataset from many different scientific perspectives in the future. Nevertheless, we expanded our Introduction and Discussion parts in our revised manuscript to further elaborate on our methodology and scientific findings, as follows.

Page 4, Line 11 in Introduction Part: "In addition, the sleep/wake states are usually accompanied by distinct neural events which result from the synchronous activities of neural circuits, for example, spindles or slow waves in NREM sleep^{21, 22} and sharp wave ripples (SWRs)^{23, 24} in both quiet wakefulness and NREM sleep. Previous studies have shown that these events play important roles in sleep architecture, synaptic plasticity and memory consolidation²⁴⁻²⁶, among many others."

Page 5, Line 2 in Introduction Part: "In addition, the brain state dynamics, as broadly

defined, has been widely investigated and implicated in cognitive processes and brain disorders¹⁶⁻¹⁹.”

Page 17, Line 1 in Discussion Part (as in response to Major point 4): “In addition, we found the co-occurrence of SWRs and spindles elicited enhanced BOLD responses, compared to the sum of the responses of two solitary events (Fig. 8 in the revised manuscript). Such effect was most notable in hippocampus, RSP and thalamus. Previous studies have shown that the interaction between thalamocortical spindles and hippocampal ripples promoted memory consolidation²¹. Interruption of the synchronization between ripple and spindle events appeared to interfere the efficiency of memory consolidation²⁷. Thus, we speculate that the synergistic effects of SWRs and spindles might be related to the memory consolidation process, and further research is needed to examine the functional relevance of such phenomenon.”

Page 17, Line 9 in Discussion Part (as in response to Major point 5): “Moreover, slow waves are also thought to be critical for initiating information transfer between hippocampus and neocortex²². A previous human EEG-fMRI study²⁸ showed significant slow wave evoked activations in right parahippocampal gyrus, precuneus and posterior cingulate cortex, which were in good accordance with our results (Now Supplementary Fig. 25b). It is known that the relationship among slow waves, spindles and SWRs is complex and may contribute to many cognition processes²⁹, especially memory consolidation^{21,27}. Thus, the relationship among these events can be further explored, potentially using the current dataset.”

Major points:

1. It is not clear how the authors performed sleep scoring. How did they score 1 s epochs? The reference they refer for established criteria is not describing any details for sleep scoring and therefore does not provide the necessary information to judge about the authors approach. Please, provide in depth information on these procedures. Especially, if sleep classification was (semi-) automated or manual. The available data clearly shows that sleep staging does not follow established criteria and must be checked very carefully.

Response:

We sincerely apologized for the lack of details in brain state classification. We followed the conventional mouse sleep state classification criteria in the mouse sleep field. The brain state classification procedure³⁰ was directly adopted from our colleagues Prof. Min Xu and Prof. Zhe Zhang (acknowledged in the manuscript), who were both trained in Prof. Yang Dan (UC Berkeley)’s lab. Now we added more information to better describe the sleep state classification procedure in Page 24 Line 27, as follows:

“For brain state classification, one channel within each session was selected for further analysis, and the selected channels were listed in the Supplementary File 6. Then, we calculated the power spectrum for the ECoG/iEEG and EMG (160-250Hz) data with 3s sliding windows and 1s step size, using the multi-taper method implemented in Chronux (<http://chronux.org/>). Next, we computed the theta (6-12 Hz) and delta (1-4 Hz) power and theta/delta power ratio, which were further smoothed using “medfilt1” (20 points) in MATLAB. For each session, we used the following criteria for tentatively defining brain states: (1) a time point was classified as NREM sleep if the smoothed delta power was higher than its mean; (2) a time point was assigned as REM sleep if the smoothed theta/delta power ratio was two standard deviations

higher than its mean and the EMG power was one standard deviation lower than its mean; and (3) all remaining time points were classified as AW state. Then, we further manually adjusted the classification of brain states as following: (1) For NREM state, according to the ECoG power spectrum, we adjusted the start or end point to the point with the greatest ascent or descent speed of smoothed delta power. For REM state, the start point was adjusted to the end point of the previous NREM, and the end point was adjusted to the point with the greatest descent speed of smoothed theta power. If the greatest ascent or descent point of EMG power was different from that of ECoG/iEEG signals, the midpoint between the two was defined as the transition point; (2) For sessions with noisy EMG recordings, we used the head motion (framewise displacement) estimated from fMRI data as substitute for EMG power; and (3) epochs with short durations (< 5s) were manually merged to the nearest sleep stage. An example of the above classification procedure (session 1: channel 10, 11500-13000s) was shown in Supplementary Fig. 3.”

Figure R1. **Example of brain states classification procedure (session 1: channel 10, 11500-13000s).** (Now supplementary Fig. 3) a Tentative brain state classification. A time point was classified as NREM sleep if the delta power was higher than its mean; a time point was assigned as REM sleep if the theta/delta power ratio was two standard deviations higher

than its mean and all remaining time points were classified as AW state. **b** Manual adjustment. For NREM state, according to the ECoG channel power spectrum, we adjusted the start or end point to the point with the greatest ascent or descent speed of smoothed delta power. For REM state, the start point was adjusted to the end point of the previous NREM, and the end point was adjusted to the point with the greatest descent speed of smoothed theta power. If the greatest ascent or descent point of EMG power was different from that of ECoG/iEEG signals, the midpoint between the two was defined as the transition point. High EMG power NREM epoch was manually excluded.

In addition, in our original manuscript, we already calculated the mean power spectrum of four state transition processes and found sharp power changes along with state transitions (Fig. R2). Such results indicated that, while it is difficult to achieve “ground truth” classification, the overall accuracy of sleep state classification was acceptable at the group level. Thus, it is unlikely to negatively impact our further analysis.

Moreover, it has been previously reported that the REM state is associated with a large increase of hemodynamic signals^{31, 32}. Similarly, we found that the BOLD fMRI signal increased markedly during REM state, which is evident at the single trial level (Fig. R3) and the group level (Fig. 3a in the original manuscript). Therefore, the above results also provide indirect evidence supporting our REM state classifications.

Figure R3. A representative session of mouse sleep fMRI with AW, NREM and REM sleep states (Mouse 26, ch5, 7400s-8400s). (Fig. 2e in the original manuscript) From upper to lower panels: power spectrogram of ECoG signal, amplitude of EMG signal, global BOLD signal and voxel-wise whole-brain BOLD signals. AW, awake; NREM, non-rapid eye movement; REM, rapid eye movement.

2. The authors often imply causality when referring to correlational analysis. E.g., line 197-200: “For REM state, significant correlation between theta band-limited power and BOLD changes was observed (Fig. 3d, e, right panels), suggesting the contribution of theta band on BOLD changes.”

Response:

We fully agree with the Reviewer’s comment. In our original manuscript, we already tried to be cautious about the interpretation of such correlational analysis. Now we have revised our manuscript to further emphasize the correlational nature more clearly:

(1) Line 197 (now Page 9 Line 16), “For REM state, significant correlation between theta band-limited power and BOLD changes was observed (Fig. 3d, e, right panels), suggesting the ~~contribution-potential relationship~~ of theta band ~~on and~~ BOLD changes.”

(2) Line 422 (now Page 18 Line 29), “Moreover, the significant correlation between electrophysiological and BOLD signals (Fig. 3d, e) ~~indicated the potential neurophysiological relevance of NREM and REM state dependent BOLD activations. suggested that the BOLD features in the current study were unlikely to be entirely vascular effects.~~”

3. (1) The authors refer to spindle band for frequency bands ranging from 7 - 37 Hz (line 260). Sleep spindles do not reach such high frequencies! This also relates to figure 6 a. (2) The authors generally should indicate the time window for which they find significant differences for these frequencies, and the exact method they used for statistical testing.

Response:

(1) We sincerely apologize for the inappropriate expression. Now, we have removed these labels of “ripple or spindle band” to avoid the misinterpretation of our results (Fig. R4) in our previous Fig. 6a.

(2) We apologize again for missing information about the time window with significant differences across frequency bands. We also added description of how the statistical testing was done. The revised part in Page 11 Line 13, as follows:

“Utilizing the GF electrode placed in hippocampus CA1 region of our LFP-fMRI dataset, we compared the LFP power spectrum between “NREM to AW” and “NREM only” states, and found significantly higher power in three frequency bands, including (1) 7-37 Hz from -14 to -2 s, (2) 44-95 Hz from -12 to -1 s, and (3) 119-205 Hz from -11 to -3 s before state transitions (Fig. 6a). Also, we conducted same analysis between “NREM to REM” and “NREM only” states, and found significantly higher power in two frequency bands, including (1) 1-34 Hz from (beyond) -30 to -3 s, and (2) 128-141 Hz from (beyond) -30 to -13 s before state transitions (Fig. 6a). These “NREM only” epochs were randomly selected from the sleep fMRI dataset, and epochs of “NREM to AW”, “NREM to REM” and “NREM only” with short durations (< 60 s) were excluded. Then, two sample t-test was conducted with threshold of $p < 0.05$.”

4. The analysis shown in figure 7 lacks the BOLD signals for solitary spindles as a control. Previous work showed that spindles are generally accompanied by a widespread increase, e.g., in cortical activity. Currently, it is not clear whether it is actually the co-occurrence of ripples and spindles that drives this wide spread increase in the BOLD signal or if spindles generally show such activity pattern. On a related note, the interpretation that coupled spindle events “contribute” to increased BOLD responses evoked by SWRs is at best unclear: BOLD activity elicited by spindles not coupled to SWRs would have to be subtracted to substantiate this claim,

and the combined events are likely more than the sum of their parts, both in terms of the neural activities elicited and the functions subserved.

Response:

We appreciate the Reviewer's constructive comments. In our original manuscript, we already provided the results of solitary spindle evoked BOLD response in Supplementary Fig. 22. Per the Reviewer's suggestions, we expanded the analysis of spindle events and ripple events, to examine whether there's any difference between BOLD responses of coupled events and summed events. Indeed we found such difference and it summarized in Fig. 8 along with the previous Supplementary Fig. 22 in our revised manuscript.

Now, we added more information in our manuscript in Page 11 Line 1 (Results part) and Page 17 Line 1 (Discussion part), as follows:

Page 11 Line 1: "To investigate whether there was any synergistic effects of the SWRs and spindles triggered BOLD responses, we firstly summed the BOLD responses of solitary spindles and SWRs in each session ("summed responses"), as well as those of spindle coupled SWRs ("coupled responses"). Then, we conducted paired t-test across sessions between the above coupled and summed responses at regional (Fig. 8b-d) and whole brain levels (Fig. 8e). We found a "coupled > summed" response across cortical regions and "coupled < summed" response in thalamus (Fig. 8b-e), suggesting synergistic effects of the SWRs and spindles triggered BOLD responses."

Page 17 Line 1: "In addition, we found the co-occurrence of SWRs and spindles elicited enhanced BOLD responses, compared to the sum of the responses of two solitary events (Fig. 8). Such effect was most notable in hippocampus, RSP and thalamus. Previous studies have shown that the interaction between thalamocortical spindles and hippocampal ripples promoted memory consolidation²¹. Interruption of the synchronization between ripple and spindle events appeared to interfere the efficiency of memory consolidation²⁷. Thus, we speculate that the synergistic effects of SWRs and spindles might be related to the memory consolidation process, and further research is needed to examine the functional relevance of such phenomenon."

Figure R5. **Synergistic effects of SWR-spindle coupling.** (Now Fig. 8) **a** Solitary spindle ($n=4288$ epochs) triggered BOLD responses in the NREM state. Results in the axial view were shown in Supplementary Fig. 23. **b-d** Significant differences between spindle coupled SWRs triggered BOLD response and the summation of BOLD responses of solitary spindles and SWRs in mPFC, HPF and Thal. Each dot represented an individual session. Statistical significance was tested using the two-tailed paired t-test. **e** Whole brain mapping of the synergistic effects of SWR-spindle coupling. Statistical significance was tested using the two-tailed paired t-test. mPFC, medial prefrontal cortex; HPF, hippocampal formation; Thal, thalamus.

5. One important omission that would allow for better embedding of the present findings into existing models of sleep is that the authors did not investigate/explore BOLD activity associated with slow oscillations which are thought to play a central role in initiating information transfer between hippocampus and neocortex.

Response:

We really appreciated the Reviewer's comments and we added slow wave evoked BOLD signal response in Supplementary Fig. 25. The procedure of identifying slow wave events was added in the Methods part in Page 25 line 30 as follows:

Page 25 line 30: "Slow waves were identified based on the procedures described previously²². The raw signal was first down-sampled to 1024 kHz. Then, for slow waves detection, mPFC iEEG signals were filtered between 0.3 and 4.5 Hz with a two-order Butterworth bandpass filter. A slow wave was detected in NREM state if the following three criteria were all fulfilled: (1) the interval (T) of negative wave between 0.4 and 2.0 s; (2) top

35% negative amplitude (N) and (3) top 45% negative-to-positive peak-to-peak amplitude (M). Slow wave onset and offset was defined by the time of the first and third zero crossing, respectively.”

Also, we added slow wave related results in our manuscript in Results part in Page 13 Line 8, and Discussion part (Page 15 Line 9), as follows:

Page 13 Line 8: “Slow waves are thought to participate in the regulation of NREM sleep process. Using the same NET-fMRI approach, we also calculated the slow wave triggered spatiotemporal map and found the BOLD activations in RSP and thalamus (Now Supplementary Fig. 25).”

Page 15 Line 9: “Moreover, slow waves are also thought to be critical for initiating information transfer between hippocampus and neocortex²². A previous human EEG-fMRI study²⁸ showed significant slow wave evoked activations in right parahippocampal gyrus, precuneus and posterior cingulate cortex, which were in good accordance with our results (Now Supplementary Fig. 25). It is known that the relationship among slow waves, spindles and SWRs is complex and may contribute to many cognition processes²⁹, especially memory consolidation^{21, 27}. Thus, the relationship among these events can be further explored, potentially using the current dataset.”

blue shade represented the identified slow wave events. Lower left panel, averaged power spectrogram and time series of slow wave events. **b** Spatiotemporal BOLD signals evoked by the slow waves (12599 epochs) in NREM state. The number of event epochs were counted under the sampling rate of 0.5 Hz (fMRI repetition time).

In general the readability of the manuscript can be improved.

Response:

We sincerely apologize for the readability issue. While the current study is indeed a complex one, we revised the manuscript throughout to improve the quality.

Minor issues:

1. - 1.160: The information contained in Suppl. Fig. 2 should be briefly spelled out here, to allow readers to grasp the basic procedures of MR preprocessing.

Response:

We sincerely apologize for not making the pre-processing clearer, thus we added more information to better describe the procedure in our manuscript (Page 8 Line 6), as follows:

“Raw fMRI data exhibited good (temporal) signal-noise-ratio with small head motion and the “6 rp + 6 Δ rp + 40 PCs” regression approach was modified from our previous studies^{14, 33} to minimize the effects of scanner drift, motion and other non-neural physiological noises (Fig. 2d, Supplementary Fig. 2). The 40 PCs were derived from fMRI signals outside the mouse brain (Supplementary Fig. 2c-d), largely capturing the non-neuronal signals¹⁵, such as head motion (Supplementary Fig. 2e), physiological effects (Supplementary Fig. 2f-g) and infra-slow drift (Supplementary Fig. 2f, and i-j). Thus, using the “6 rp + 6 Δ rp + 40 PCs” regression approach, arousal induced non-neuronal nuisance effects on fMRI signals were minimized.”

2. - 1.182: The argument that transitions between sleep stages occur more frequently is valid, but it over-emphasizes the notion of sleep stages as stationary, potentially at the cost of dynamic microstructure (which the authors do investigate with NET-fMRI).

Response:

We fully agree that the sleep stages are non-stationary, and following such notion we have investigated the low dimensional dynamics within brain states (Fig. R7) besides the NET-fMRI analysis. In addition, unlike the N1, N2 and N3 divisions of human NREM state, there is no established criteria of further division of mouse NREM sleep. Therefore, mouse NREM sleep was not further divided into 3 stages. More features of dynamic microstructure within mouse sleep states can be explored in future studies. Here we revised our manuscript (Page 15 Line 25) to further emphasize the non-stationary features of brain states, as follows:

“Hence, PCA enables analysis of the state space trajectory (or flow), which reflects the temporal evolution of the global brain state³⁴. Combined with rich resources and tools in mice, more features of dynamic microstructure within mouse sleep states can be explored in future studies. Particularly, the asymmetry trajectories in both electrophysiological and BOLD spaces indicated the asymmetrical between “AW to NREM” and “NREM to AW” transitions (Fig. 4h), further suggesting different neural circuits for awake-promoting and NREM sleep-promoting processes.”

Furthermore, we revised the wording of the results to avoid misinterpretation, as follows:

1) Line 110 (now Page 6 Line 4), “With this mouse sleep fMRI method, we revealed **static** global patterns of NREM and REM sleep, and more importantly, the global, irreversible and sequential dynamics of state transitions, which was also evident in their trajectories in the low-dimensional manifold.”

2) Line 202 (now Page 9 Line 21), “The above NREM and REM activation patterns were **static** state-dependent features.”

Figure R7. **Low dimensional dynamics within and between brain states (Fig. 4c, g, e in the original manuscript)** **a** Circular distribution of temporal weights of PCs (tPCs) from the start to the end of each 952 brain state: AW (left, n=2703 epochs), NREM (middle, n= 2498 epochs) and REM (right, n= 165953 epochs). Epochs with short durations (< 30 s) of each state were excluded and further applied to the following analysis. Radius of radar plots, tPCs; Color line, mean tPCs; gray shadow, the 95% confidence interval (CI) of null control (1000 shuffling). **b** Low dimensional manifold of BOLD signals traversed by the brain state across first three PCs, with arrows depicting the directions of flow along the manifold. **c** Similar to (b) but in two-dimensional electrophysiological space. Color dots, states per second.

3. - 1.226: I found the term “non-reversible transitions” used here and elsewhere unclear – perhaps ‘non-commutative’ or ‘asymmetric’ would be better.

Response:

We appreciated this suggestion and now replaced the term in our manuscript as follows:

Page 3 Line 13: “State dependent global patterns were revealed, and state transitions were found to be global, **irreversible asymmetric** and sequential phenomenon, which can be predicted up to 17.8s using LSTM RNN models.”

Page 6 Line 4: “With this mouse sleep fMRI method, we revealed global patterns of NREM and REM sleep, and more importantly, the global, **irreversible asymmetric** and sequential dynamics of state transitions, which was also evident in their trajectories in the low-dimensional manifold.”

Page 10 Line 12: “And we further quantified such phenomenon and the significant asymmetry were observed around state transitions in both BOLD and electrophysiological space (Fig. 4h), suggesting “AW to NREM” and “NREM to AW” transitions were **not-reversible asymmetric** processes.”

4. - 1.517: As far as I see, the habituation procedure has not been explicitly described before.

Response:

The corresponding description in Methods parts (Page 22, Line 14) was revised and expanded as follows:

Page 22 Line 14: “After seven-day recovery, mice were then habituated for fMRI for another seven days. Mice were head fixed on the animal bed with the recorded acoustic MRI scanning noise based on previous work³⁵. Optical imaging studies usually record 3-5h to obtain mice sleep more efficiently³⁶⁻³⁸, as head fixed mice frequently fall asleep after 1.5h and REM sleep frequently occurs after 2.5h³⁶. Therefore, 4-h head restraining was also utilized in our mouse sleep fMRI. The 30° head holder was designed to mimic the natural sleep gesture of mice³⁷. The animal bed was modified to allow more space for forelimb movement to reduce stress level³⁵. The habituations were all carried out during 9a.m.-15p.m. with a fixed duration of 4h and gradually increased noise levels. The detailed schedule was listed in Table 1. No reward was given during or after the habituation training.”

5. - 1.530: Why did the authors apply the wide notch filter between 100 and 150 Hz?

Response:

We sincerely apologized for the misunderstanding due to our informal description and now revised the text from “at 50/100-150 Hz” to “at 50, 100 and 150 Hz” in our manuscript (Page 23 Line 7).

6. - 1.546: In humans, the most problematic artifact with simultaneous EEG-fMRI is not due to imaging, but to pulse-induced electrode movements in the static magnetic field. I imagine these movements are considerably smaller in the present case. Nevertheless, the authors should mention if and how this potential source of artifacts was taken into account.

Response:

We really appreciated the Reviewer’s suggestions. Compared with 60-100 beats per minute in humans, the heart rate of the unanesthetized mouse varies between 300 and 600 beats per minute. However, in our dataset, after our electrophysiological denoising procedure (Now Supplementary Fig. 1), we did not observe typical pulse artifact in the 5-10 Hz frequency band from the ECoG and LFP power spectrums (Fig. R8).

This phenomenon might attribute to our intracranial placement of electrodes and the stable electrode fixation. In human EEG-fMRI studies, pulse artifact occurs when an EEG electrode is placed over a pulsating vessel³⁹. The pulsation can cause slow electrode movements that contaminate EEG activities. Our intracranial electrodes were tightly fixed on the mouse skull

by dental cement. Thus, scalp pulse and cardiac-related motion were less likely to impact the electrode movement and further influence our electrophysiological signals. Now we added more information in our manuscript (Page 24 Line 15) as follows:

“Interestingly, we did not observe typical pulse artifact from the ECoG and LFP power spectrums (Fig. 2e and Fig. 4e), which is often problematic in human EEG-fMRI studies³⁹. After our electrophysiological denoising procedure, no notable ballistocardiogram artifact was observed in 5-10 Hz frequency band, corresponding to the unanesthetized mouse heart rate between 300 and 600 beats per minute. This phenomenon might attribute to our intracranial placement of electrodes and the stable electrode fixation. In human EEG-fMRI studies, pulse artifact occurs when an EEG electrode is placed over a pulsating vessel³⁹. The pulsation can cause slow electrode movements that contaminate EEG activities. Our intracranial electrodes were tightly fixed on the mouse skull by dental cement. Thus, scalp pulse and cardiac-related motion were less likely to impact the electrode movement and further influence our electrophysiological signals.”

7. - 1.557: The authors should clarify the range and mean number of electrodes interpolated across animals. It should also be clarified how many of these were located at the border of the ECoG arrays, as interpolation results are likely less reliable in these cases.

Response:

We really appreciated the Reviewer's comments. As suggested, we provided a Supplementary File 5 to describe the interpolated channels for each session. Combined with the location and design of ECoG array (Fig. R9), readers can conveniently infer the position of interpolated electrodes.

8. - 1.609: Is there any literature on potential differences between HRFs in the mouse across different brain states?

Response:

We really appreciated the Reviewer's comment for pointing out the potential differences of HRFs across different brain states. With known variability of HRFs across species⁴⁰ brain regions³³ or even physiological conditions⁴¹, HRF differences across mouse sleep states have not been reported, to the best of our knowledge. Nevertheless, compared to the plausible HRF ranges (typically 2-5 s), long state durations (Median > 24 s, Now Supplementary Fig. 4) could improve the tolerance of the variability of HRFs and corresponding activation results. Therefore, we believed that the potential variability of HRFs is unlikely to influence our NREM and REM state dependent BOLD activations. Except the results in Fig.3, all remaining analysis did not involve the HRFs.

9. - 1.659: How did the authors determine the cut-off of 100 temporal principal components?

Response:

We sincerely apologize for not describing the rationale for determining the PCA number. Now the revised Supplementary Fig. 6a showed the explained variances across all 7199 PCs (Fig. R10). (The number of repetitions was 7200 for each fMRI EPI run, so 7199 is the

maximum.) We found the 100th PC accounted for at least 0.3% explained variances and accumulatively the first 100 PCs resolved greater than 78% explained variances of BOLD signals. Now, we have revised our manuscript in Page 9 line 24, as follows:

“First, we conducted group principal component analysis (PCA) to BOLD signals for dimensional reduction. The -100th PC accounted for at least 0.3% explained variances and accumulatively the first 100 PCs resolved greater than 78% explained variances of BOLD signals. (Fig. 4a, Supplementary Fig. 6, and Supplementary File 2).”

- Out of curiosity: can the authors say anything about spindle-associated cerebellar activity? (cf. Xu, W., De Carvalho, F., Clarke, A. K., & Jackson, A., 2021, Progress in Neurobiology)

Response:

Unfortunately, our field of view (FOV) of fMRI data (Fig. 1a) didn't cover the cerebellar areas. Therefore, it is not possible to investigate the spindle-associated cerebellar activity using our dataset. In the future, further efforts can be made to provide cerebellum coverage.

Reference:

1. McGinley MJ, David SV, McCormick DA. Cortical Membrane Potential Signature of Optimal States for Sensory Signal Detection. *Neuron*. 2015;87(1):179-192.
2. Gu Y, Han F, Liu X. Arousal Contributions to Resting-State fMRI Connectivity and Dynamics. *Front Neurosci*. 2019;13:1190.
3. He BJ, Snyder AZ, Zempel JM, et al. Electrophysiological correlates of the brain's intrinsic large-scale functional architecture. *Proc Natl Acad Sci U S A*. 2008;105(41):16039-16044.
4. Raut RV, Snyder AZ, Mitra A, et al. Global waves synchronize the brain's functional systems with fluctuating arousal. *Sci Adv*. 2021;7(30).
5. Mitra A, Kraft A, Wright P, et al. Spontaneous Infra-slow Brain Activity Has Unique Spatiotemporal Dynamics and Laminar Structure. *Neuron*. 2018;98(2):297-305.e296.
6. Mitra A, Snyder AZ, Tagliazucchi E, et al. Propagated infra-slow intrinsic brain activity reorganizes across wake and slow wave sleep. *Elife*. 2015;4.
7. Palva JM, Palva S. Infra-slow fluctuations in electrophysiological recordings, blood-oxygenation-level-dependent signals, and psychophysical time series. *Neuroimage*. 2012;62(4):2201-2211.
8. Chang C, Leopold DA, Schölvinck ML, et al. Tracking brain arousal fluctuations with fMRI. *Proc Natl Acad Sci U S A*. 2016;113(16):4518-4523.
9. Oken BS, Salinsky MC, Elsas SM. Vigilance, alertness, or sustained attention: physiological basis and measurement. *Clin Neurophysiol*. 2006;117(9):1885-1901.
10. Liu X, de Zwart JA, Schölvinck ML, et al. Subcortical evidence for a contribution of arousal to fMRI studies of brain activity. *Nat Commun*. 2018;9(1):395.
11. Salimi-Khorshidi G, Douaud G, Beckmann CF, et al. Automatic denoising of functional MRI data: combining independent component analysis and hierarchical fusion of classifiers. *Neuroimage*. 2014;90:449-468.
12. Glover GH, Li TQ, Ress D. Image-based method for retrospective correction of physiological motion effects in fMRI: RETROICOR. *Magn Reson Med*. 2000;44(1):162-167.
13. Han Z, Chen W, Chen X, et al. Awake and behaving mouse fMRI during Go/No-Go task. *Neuroimage*. 2019;188:733-742.
14. Tong C, Liu C, Zhang K, et al. Multimodal analysis demonstrating the shaping of functional gradients in the marmoset brain. *Nat Commun*. 2022;13(1):6584.
15. Chuang KH, Lee HL, Li Z, et al. Evaluation of nuisance removal for functional MRI of rodent brain. *Neuroimage*. 2019;188:694-709.
16. Fox MD, Snyder AZ, Barch DM, et al. Transient BOLD responses at block transitions. *Neuroimage*. 2005;28(4):956-966.
17. Hayden BY, Smith DV, Platt ML. Electrophysiological correlates of default-mode processing in macaque posterior cingulate cortex. *Proc Natl Acad Sci U S A*. 2009;106(14):5948-5953.
18. Lustig C, Snyder AZ, Bhakta M, et al. Functional deactivations: change with age and dementia of the Alzheimer type. *Proc Natl Acad Sci U S A*. 2003;100(24):14504-14509.
19. Luo L, Li Q, You W, et al. Altered brain functional network dynamics in obsessive-compulsive disorder. *Hum Brain Mapp*. 2021;42(7):2061-2076.
20. Barttfeld P, Uhrig L, Sitt JD, et al. Signature of consciousness in the dynamics of resting-state brain activity. *Proc Natl Acad Sci U S A*. 2015;112(3):887-892.
21. Ngo HV, Fell J, Staresina B. Sleep spindles mediate hippocampal-neocortical coupling during

long-duration ripples. *Elife*. 2020;9.

22. Oyanedel CN, Durán E, Niethard N, et al. Temporal associations between sleep slow oscillations, spindles and ripples. *Eur J Neurosci*. 2020;52(12):4762-4778.

23. Buzsáki G. Hippocampal sharp waves: their origin and significance. *Brain Res*. 1986;398(2):242-252.

24. Buzsáki G. Hippocampal sharp wave-ripple: A cognitive biomarker for episodic memory and planning. *Hippocampus*. 2015;25(10):1073-1188.

25. Fernandez LMJ, Lüthi A. Sleep Spindles: Mechanisms and Functions. *Physiol Rev*. 2020;100(2):805-868.

26. Adamantidis AR, Gutierrez Herrera C, Gent TC. Oscillating circuitries in the sleeping brain. *Nat Rev Neurosci*. 2019;20(12):746-762.

27. Novitskaya Y, Sara SJ, Logothetis NK, et al. Ripple-triggered stimulation of the locus coeruleus during post-learning sleep disrupts ripple/spindle coupling and impairs memory consolidation. *Learn Mem*. 2016;23(5):238-248.

28. Dang-Vu TT, Schabus M, Desseilles M, et al. Spontaneous neural activity during human slow wave sleep. *Proc Natl Acad Sci U S A*. 2008;105(39):15160-15165.

29. Grimaldi D, Papalambros NA, Zee PC, et al. Neurostimulation techniques to enhance sleep and improve cognition in aging. *Neurobiol Dis*. 2020;141:104865.

30. Zhong P, Zhang Z, Barger Z, et al. Control of Non-REM Sleep by Midbrain Neurotensinergic Neurons. *Neuron*. 2019;104(4):795-809.e796.

31. Bergel A, Deffieux T, Demene C, et al. Local hippocampal fast gamma rhythms precede brain-wide hyperemic patterns during spontaneous rodent REM sleep. *Nat Commun*. 2018;9(1):5364.

32. Turner KL, Gheres KW, Proctor EA, et al. Neurovascular coupling and bilateral connectivity during NREM and REM sleep. *Elife*. 2020;9.

33. Chen X, Tong C, Han Z, et al. Sensory evoked fMRI paradigms in awake mice. *Neuroimage*. 2020;204:116242.

34. Shine JM, Breakspear M, Bell PT, et al. Human cognition involves the dynamic integration of neural activity and neuromodulatory systems. *Nat Neurosci*. 2019;22(2):289-296.

35. Xu W, Pei M, Zhang K, et al. A systematically optimized awake mouse fMRI paradigm. *bioRxiv*. 2022:2022.2011.2016.516376.

36. Dong Y, Li J, Zhou M, et al. Cortical regulation of two-stage rapid eye movement sleep. *Nat Neurosci*. 2022;25(12):1675-1682.

37. Yuzgec O, Prsa M, Zimmermann R, et al. Pupil Size Coupling to Cortical States Protects the Stability of Deep Sleep via Parasympathetic Modulation. *Curr Biol*. 2018;28(3):392-400 e393.

38. Xie L, Kang H, Xu Q, et al. Sleep drives metabolite clearance from the adult brain. *Science*. 2013;342(6156):373-377.

39. Bullock M, Jackson GD, Abbott DF. Artifact Reduction in Simultaneous EEG-fMRI: A Systematic Review of Methods and Contemporary Usage. *Front Neurol*. 2021;12:622719.

40. Tong C, Dai JK, Chen Y, et al. Differential coupling between subcortical calcium and BOLD signals during evoked and resting state through simultaneous calcium fiber photometry and fMRI. *Neuroimage*. 2019;200:405-413.

41. Chao TH, Zhang WT, Hsu LM, et al. Computing hemodynamic response functions from concurrent spectral fiber-photometry and fMRI data. *Neurophotonics*. 2022;9(3):032205.

REVIEWERS' COMMENTS

Reviewer #1 (Remarks to the Author):

I am very pleased by the thoughtful responses from the authors to my concerns. I enthusiastically support the publication of this very important study which I predict will be welcomed by the neuroscience research community interested in sleep. The strong commitment of these authors to share their data is an added exemplary feature.

Reviewer #2 (Remarks to the Author):

While the authors sufficiently addressed most of my previous concerns, I have to reiterate that the paper in its current form does not sufficiently embed the authors' findings into the field of sleep research. The current manuscript somewhat is a mixture between a dataset report, a methods paper and a research report. The findings are indeed interesting and therefore should sufficiently be discussed in relation to what is known about circuit activity during wakefulness and sleep.

In particular, the authors widely ignore literature that describes changes in neuronal activity during brain stage transitions and sleep specific oscillations using e.g. electrophysiological recordings or calcium imaging.

In addition, the authors should clearly state the limitations of their automated sleep scoring approach especially because sleep recordings under headfixation differ substantially from those in freely moving animals.

REVIEWER COMMENTS

Reviewer #2 (Remarks to the Author):

Nature Communications manuscript NCOMM-22-50604-T

Title: Mouse sleep fMRI with simultaneous electrophysiology

While the authors sufficiently addressed most of my previous concerns, I have to reiterate that the paper in its current form does not sufficiently embed the authors' findings into the field of sleep research. The current manuscript somewhat is a mixture between a dataset report, a methods paper and a research report. The findings are indeed interesting and therefore should sufficiently be discussed in relation to what is known about circuit activity during wakefulness and sleep.

In particular, the authors widely ignore literature that describes changes in neuronal activity during brain stage transitions and sleep specific oscillations using e.g. electrophysiological recordings or calcium imaging.

Response:

We sincerely appreciate the comments of reviewers. We had discussed several related circuitry level studies, e.g., Page 16 from line 15 to 32 for prediction of state transitions, from Page 17 line 30 to Page 18 line 6 for SWRs/spindles triggered global pattern and Page 16 from line 2 to 8 for dynamics within brain states. Now we further expanded such discussion as follows:

Page 17 line 2: "Sleep-wake cycle is believed to be tightly regulated by a distributed network of sleep and wake promoting neurons^{1,2}, primarily located in subcortical regions. Recently, cortical regulation of sleep transition and maintenance has also been reported. Silencing neocortical layer 5 pyramidal neurons using SNAP25 knockout mouse decreased the cortico-subcortical communication and further markedly increased wakefulness³. In our study, we found widespread changes of cortical activities during "AW to NREM" and "NREM to AW" state transitions (Supplementary Fig. 7a, b), and some changes occurred earlier than those in subcortical regions, highlighting the important role of the cortex. Another notable example of cortical involvement in sleep regulation is retrosplenial cortex (RSP). Two recent studies in mice both showed that RSP was critically involved in REM sleep initiation and progression^{4,5}. The current study also found RSP was highly activated during both NREM and REM state, and the activity of posterior DMN-like network (PC4 in Fig. 4a and e), largely overlapping with RSP, apparently preceded the "NREM to REM" state transition. Therefore, our study agrees well with recent progress on the cortical involvement in sleep-wake transitions."

Page 17 line 24: "Activation in RSP and hippocampus and deactivations in thalamus were consistent with previous results in macaque⁶. And optical imaging during natural sleep and anesthesia in mice also showed significant activation of RSP during SWRs⁷. Converging evidence, including the current results, clearly suggests the important role of RSP during SWRs, which may be related to the critical role of subiculum-retrosplenial pathway for SWR propagation from hippocampus to neocortex⁸."

In addition, the authors should clearly state the limitations of their automated sleep scoring

approach especially because sleep recordings under headfixation differ substantially from those in freely moving animals.

Response:

Thank you for your insightful advice. Now we added more Discussions about our automated sleep scoring approach limitations (Page 19 Line 18), as follows:

Page 19 Line 18: “Then, the sleep scoring criteria employed in the current study was largely adopted from previous mouse sleep studies⁹ in the free-moving state. As the current study was the first to develop mouse sleep fMRI and thus no prior knowledge about ECoG/LFP and EMG signal characteristics during mouse sleep fMRI was known, the semi-automated sleep scoring approach developed here may be further refined. While we designed the 30° head holder to mimic the natural posture, sleep in the natural free-moving state may still be different from that in the head fixed state, which may result in different ECoG/LFP and EMG signal characteristics. Although other optical imaging studies of mouse sleep conducted in the head fixed condition also employed similar sleep scoring criteria^{5,10}, further detailed examination is much needed for improving sleep scoring in the head fixed state.”

Reference:

- 1 Liu, D. & Dan, Y. A Motor Theory of Sleep-Wake Control: Arousal-Action Circuit. *Annu Rev Neurosci* **42**, 27-46, doi:10.1146/annurev-neuro-080317-061813 (2019).
- 2 Scammell, T. E., Arrigoni, E. & Lipton, J. O. Neural Circuitry of Wakefulness and Sleep. *Neuron* **93**, 747-765, doi:10.1016/j.neuron.2017.01.014 (2017).
- 3 Krone, L. B. *et al.* A role for the cortex in sleep-wake regulation. *Nat Neurosci* **24**, 1210-1215, doi:10.1038/s41593-021-00894-6 (2021).
- 4 Dong, Y., Li, J., Zhou, M., Du, Y. & Liu, D. Cortical regulation of two-stage rapid eye movement sleep. *Nat Neurosci* **25**, 1675-1682, doi:10.1038/s41593-022-01195-2 (2022).
- 5 Wang, Z. *et al.* REM sleep is associated with distinct global cortical dynamics and controlled by occipital cortex. *Nat Commun* **13**, 6896, doi:10.1038/s41467-022-34720-9 (2022).
- 6 Logothetis, N. K. *et al.* Hippocampal-cortical interaction during periods of subcortical silence. *Nature* **491**, 547-553, doi:10.1038/nature11618 (2012).
- 7 Karimi Abadchi, J. *et al.* Spatiotemporal patterns of neocortical activity around hippocampal sharp-wave ripples. *Elife* **9**, doi:10.7554/eLife.51972 (2020).
- 8 Nitzan, N. *et al.* Propagation of hippocampal ripples to the neocortex by way of a subiculum-retrosplenial pathway. *Nat Commun* **11**, 1947, doi:10.1038/s41467-020-15787-8 (2020).
- 9 Zhong, P. *et al.* Control of Non-REM Sleep by Midbrain Neurotensinergic Neurons. *Neuron* **104**, 795-809.e796, doi:10.1016/j.neuron.2019.08.026 (2019).
- 10 Turner, K. L., Gheres, K. W., Proctor, E. A. & Drew, P. J. Neurovascular coupling and bilateral connectivity during NREM and REM sleep. *Elife* **9**, doi:10.7554/eLife.62071 (2020).